# Reclassifying lethal heat

Robert Edwin Rouse [1,10] ✉, Ramit Debnath [2,3], David Andrew Rouse[4], Will Tebbutt [5,11], Bénédicte Dousset [6], J. Scott Hosking [7,8], Allan McRobie [5], Camilo Mora [9] & Emily Shuckburgh [1]

As heatwaves increase in both frequency and intensity globally, the need to develop tools to predict the human impact and develop a more comprehensive understanding of the impact mechanism at a population level is becoming more urgent. Our study provides a taxonomy of heatwaves based on identifying sub-threshold lethal heatwaves through physiological adaptation and vulnerability. We use a classification algorithm applied to a lethal heatwave dataset, comprising 125,411 events where the temperature exceeded the 90th percentile across 140 cities, with combined meteorology and socio-demographic inputs to label these events. The accuracy of our model outperforms classification that relies on wet bulb temperature thresholds with a factor of 11 improvement in imbalanced classification performance. Furthermore, we find that the majority of lethal heatwaves within our dataset occur below high wet bulb temperature thresholds and that accurate predictions for heatwave mortality can be obtained by combining thermo-temporal differentials and population health metrics instead of absolute climatic conditions. We thus propose classifying heatwaves as either: Shock Heatwaves, where aggressive thermo-temporal differentials from a local acclimation point trigger adverse stress effects, particularly among the vulnerable; or Threshold Heatwaves, where high temperature and humidity conditions do exceed the ability to dissipate heat effectively.

Anthropogenic climate change is increasing the intensity and frequency of heatwaves globally, which in turn is driving an increase in heat-related deaths and hospitalisations. For example, 37% of warm-season heat-related mortality in 43 countries has been attributed to anthropogenic forcing[1]. Infants, women, people facing socio-economic hardship and adults above the age of 65 are particularly at risk and vulnerable to heat-related health impacts[2]. Several studies have attempted to connect the biophysical, social and epidemiological interactions associated with heatwaves, but a comprehensive understanding of what makes a heatwave lethal at the interactions of sociological and physiological factors (i.e., human-level factors) is missing[3].

Climate scientists typically define heatwaves as prolonged episodes of abnormally high temperatures and rely on 'environmental stressors' like humidity, wind speed and solar radiation to derive multiple heatwave indicators[4]. At the moment, however, many studies that look for and attribute deadly heatwaves are primarily based on those definitions and these studies rarely go one step further and determine how vulnerable people are physically and socially contextualised in the definitions of 'lethal' heatwaves[1,3]. Growing number of epidemiological studies have begun to link population mortality and morbidity outcomes between socially vulnerable groups on an aggregated scale with a common conclusion among them of the need

[1]Department of Computer Science and Technology, University of Cambridge, Cambridge, UK. [2]Collective Intelligence & Design Group and climaTRACES lab, University of Cambridge, Cambridge, UK. [3]Climate and Social Intelligence Lab, California Institute of Technology, Pasadena, CA, USA. [4]Home Office Pathologist (Retired), Home Office, London, UK. [5]Department of Engineering, University of Cambridge, Cambridge, UK. [6]Hawai'i Institute of Geophysics and Planetology, University of Hawai'i at Mānoa, Honolulu, HI, USA. [7]British Antarctic Survey, Cambridge, UK. [8]Alan Turing Institute, London, UK. [9]Department of Geography and Environment, University of Hawai'i at Mānoa, Honolulu, HI, USA. [10]Present address: Department of Applied Mathematics & Theoretical Physics, University of Cambridge, Cambridge, UK. [11]Present address: Alan Turing Institute, London, UK. ✉e-mail: rer44@cam.ac.uk

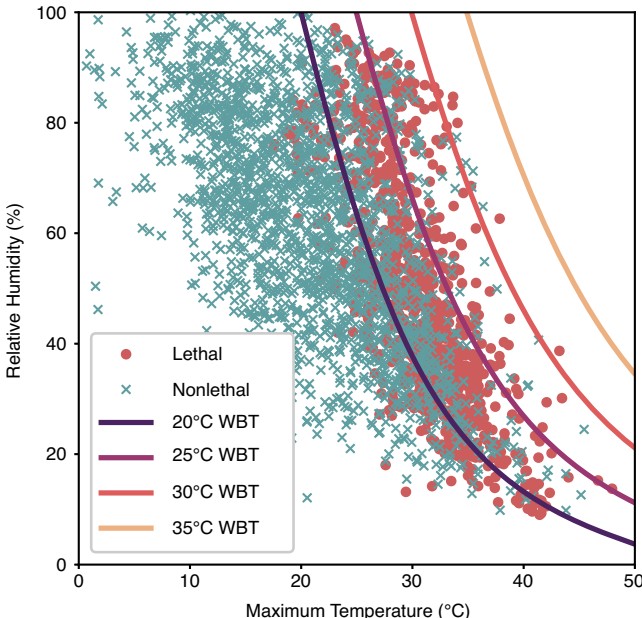

**Fig. 1 | Heatwave lethality compared with wet bulb temperature thresholds.** All lethal and a 2% subset of nonlethal heatwaves, indicated by the red circles and blue crosses respectively, as a function of maximum temperature and mean humidity during the heatwave with 20 °C, 25 °C, 30 °C and 35 °C wet bulb temperature (WBT) threshold boundaries.

for more granular attribution between heatwave types and their health outcomes[3,5–9].

Heat-related mortality data continue to remain sparse and, although there have been recent studies beginning to systematically analyse them, the definition or classification of what makes a heatwave lethal remains inconsistent[10–15]. At a physiological level, many challenges have hampered the accurate assessment of the risk of heat-related mortality. Strict hyperthermia with heat-related denaturation of body tissues may not occur as such in temperate countries or may be misdiagnosed[16,17]. The effect is more insidious, with an increased risk of fatal thrombotic events[18] or overload of already failing hearts due to the increased requirements of cutaneous blood flow, or attribution to previous comorbidities, such as diabetes, psychotic illness and various medications that interfere with thermoregulation.

We do not dispute that an absolute upper boundary of human thermotolerance exists that can be expressed in terms of temperature and humidity[19–21], but, when we considered instances of lethal heatwaves within the United Kingdom, there was an increase in excess mortality for heat events at temperatures below those in typically warmer locations and even for routinely experienced summer periods[10,15,22,23]. The issue is that this thermotolerance limit cannot be comparatively low, such as a wet bulb temperature of 25 °C, otherwise, there would be higher instance of heat-related mortality than reported at many locations[10]; conversely, if it is comparatively high, as suggested in some literature[19,20], at a threshold of circa 35 °C, then that alone would not explain the fatalities that we observed in cooler climates during relative extreme heat phenomena.

Given that wet bulb temperature thresholds of various description are often used to define a decision boundary over whether or not heatwaves are lethal and often have to be tailored geographically[24–26], these frameworks lack suitable generalising capability. In cooler, temperate countries, utilising an absolute upper boundary of human thermotolerance would result in an underprediction of mortality. However, the use of absolute threshold classifiers and warning systems persists; for example, the UK Health Security Agency relies on absolute temperature alone to inform its weather-health alert system[27]. Instead,

we prefer to consider the limits of acclimatisation or elastic thermo-tolerance capability in driving vulnerability to lethal heat or, more concretely, the temperature that a population has adapted to and its capability to deal with the stress caused by both minor and major changes in temperature in driving heatwave mortality. Therefore, we aim to create a framework that uses an empirically derived decision boundary based on the absolute conditions and the vulnerability and physiological adaptability of a given population. We hope that it is more universally applicable and sensitive to the risks of that population[3,28] and that it can better predict lethal heatwaves subject to those risks. We do this by building upon work done by Mora et al., in which they trained a Support Vector Machine to identify lethal climatic conditions before using that model to project instance of deadly heat under future climate scenarios[10]. The model was trained on a dataset of lethal and nonlethal heatwaves, which was compiled through the authors analysing 30,000 peer-reviewed sources to identify lethal and nonlethal heatwaves. The criteria for lethality or nonlethality was whether or not contemporaneous or displaced fatalities as a direct consequence of a heatwave were reported within the evidence base.

In this work, we take the dataset of heatwave events that exceed the 90th temperature percentile, with respect to historical conditions and that comprises 125,411 heatwave events occurring in 140 cities from 36 countries around the world. There are 979 lethal heatwaves with the rest being nonlethal; as a percentage, 0.78% are lethal and 99.2% are nonlethal, resulting in a highly imbalanced dataset. Although the severity of these events varies in terms of the total mortality, such as 4867 excess deaths in Paris arising from the 2003 heatwave[29] and 11 for the 1999 heatwave that affected Milwaukee[30], our goal is to establish a more appropriate decision boundary for likelihood of a heatwave being lethal based on the incompleteness of existing boundaries. We also note that this dataset is not exhaustive and not all historical lethal events for a given location are necessarily included, such as is the case for India[28].

We then use that dataset to develop a machine learning model that better predicts the probability of a heatwave being lethal by incorporating variables that can enable the model to infer population vulnerability, remedying the underprediction of mortality by relying solely on wet bulb temperatures or similar. This is achieved by combining rates of temperature change, when compared to local averages preceding each of the heatwaves, the absolute conditions of the heatwaves and national sociodemographic data.

Our model outputs a probability of a heatwave being lethal with a probability threshold being used to determine final classification. We show that our classifier outperforms the use of a similar wet bulb temperature model by at least an order of magnitude, when accounting for the issue of class imbalance. Through feature analysis and investigating subsets of the input variables, we show that a classifier can be trained without any information on the absolute conditions of a heatwave, using variables that account for vulnerability and adaptation along with the relative deviation from acclimated conditions, and again achieve nearly an order of magnitude improvement on a wet bulb temperature classification. Consequently, we hope our model might increase understanding of and responses to heatwave related mortality, in and outside of tropical or other high mean temperature zones, under future climate and health scenarios and guide the creation of mitigation and adaptation strategies. Finally, through our development of this more robust framework, we propose a taxonomy of lethal heatwaves for practitioners working within and across a broad range of climates experiencing heatwaves with different climatic characteristics.

## Results
We first aimed to address the completeness of temperature and humidity as the dependent variables for heatwave lethality to underscore the rationale for including more information. In Fig. 1, the lethal

events and a randomly selected subset of 2% of the nonlethal events, to improve visibility, are plotted in the temperature-humidity plane, as otherwise the lethal heatwaves would be obscured by the non-lethal due to the overlap.

We can observe that there is a large degree of overlap between lethal and non-lethal heatwaves and conclude that heatwave lethality cannot be determined from maximum temperature and relative humidity alone. To further this point, if we take wet bulb temperature, using the formulation from Stull[31] and derive boundaries at 35 °C, 30 °C, 25 °C and 20 °C, then we can show that none of the catalogued lethal heatwaves occurred at a wet bulb temperature of more than the commonly used limit of human adaptability of 35 °C[19,20]. In fact, 56.5% of the lethal heatwave events (=554) occurred below a wet bulb temperature of 25 °C; of the nonlethal events, 12.8% (=15,968) occurred over the 25 °C wet bulb temperature, with 8.4% (=1343) over 30 °C. In other words, more non-lethal heatwaves were documented above that 30 °C threshold than there were lethal heatwaves documented within this dataset altogether. As a result, the probability of a heatwave being lethal given a wet bulb temperature of 25 °C is less than 6% and given a wet bulb temperature of 30 °C is less than 4%. Whilst a wet bulb temperature threshold of 35 °C has already been found to have limitations[21], temperature and humidity in any combination are likely insufficient predictors and there is other behaviour at play beyond the instantaneous meteorological that must be captured.

We assume that we need additional predictors that can account for the mortality at low temperature, low humidity events. Our hypothesis is that the majority of the events recorded as lethal cause fatalities due to permanent and transient vulnerabilities within the population, either physiological and/or sociological. However, comprehensive diagnosis of all underlying comorbidities and direct measurement of physiological adaptation are not available for entire populations within all global subdomains (whether that subdomain is taken to be at the city, county, state, or national level) and that is likely to remain the case for some time.

Therefore, we take a step back and consider what data are available and that can, crucially, allow us to infer the physiological vulnerability and adaptive capacity of a population. We assume that the ability of an individual to adapt to new temperatures and humidities, in terms of rate of change and capacity, is dependent on the healthy function of physiological systems, such as the cardiovascular and endocrine systems. Further assuming that public health and sociodemographic data contain some quantity of information regarding the physiological function of a population, such data ought to contain information about a population's adaptive capability. Consequently, we assume at any given point in time the adaptation to that instance's meteorological state is dependent on the underlying physiological adaptive capacity of a population and the antecedent meteorological conditions, particularly in terms of rate of change. To formalise this, let $\rho$ denote the underlying health of a population, $c$ the antecedent climate conditions and $h$ the heatwave conditions. We assume that the heatwave impact, $H$, is characterised by some function, $f(\rho, c, h)$, which we shall infer from data and which internalises the physiological adaptation to a given set of thermal conditions.

To address the first part of that equation, we utilise the mean Body Mass Index (BMI) score for each country[32] and population pyramid data from the UN[33]. We acknowledge that BMI is an imperfect measure but, given the association of a range of non-overlapping diseases with obesity as defined by BMI[34–36], we believe it to be suitable at a population level. To compress the population pyramid data into fewer variables, we take the weighted average of the age and fit a line to the widths of the pyramid segments by linear regression, taking the gradient. We note that highly concave or convex population pyramids might be problematic for this approach. Additionally, we have utilised the Global Burden of Disease Study 2021 Socio-Demographic Index[37] mean, which estimates the burden of diseases, injuries and risk factors

for a given country by creating a composite indicator of status that was found to be strongly correlated with health outcomes. SDI is calculated from the geometric mean of indices of total fertility rate under the age of 25, mean education for those ages 15 and older and lag distributed income per capita. The sociodemographic and health data we use must match or extend beyond the temporal range of the lethal heatwave data, starting in or before 1980 and continuing past 2014. Consequently, potentially useful datasets, such as the GLOPOP-S dataset that details characteristics that drive different behavioural responses to environmental risk[38], cannot be used as they do not match the required temporal range. Including such data would effectively result in data poisoning and likely negatively impact the model, precluding the use of these other datasets.

To address the second part, the antecedent climate conditions, we calculate the temperature differential from the heatwave maximum to preceding period averages, namely the 30, 90 and 180 day periods, which we denote as $\Delta T_{30}$, $\Delta T_{90}$ and $\Delta T_{180}$, respectively. These variables, having different temporal ranges, capture local temperature seasonality along with the relative intensity of the heatwave over recent conditions. We also introduce an additional antecedent variable to capture an element of sociological adaptation, or, conversely, how unusual it is for a society to experience a given temperature with respect to seasonality. Over a sufficiently long period of time, one might reasonably assume that a society has implemented accessible measures on a broad enough scale to help its members cope with high temperatures, such as air conditioning, or adopted certain behaviours, such as a shift in working patterns around the hottest part of the day. We refer to this as adaptive temperature. We have also generated a similar set of variables for humidity assuming that there might be some physiological and sociological adaptation to wet bulb conditions beyond ordinary temperature. These variables are $\Delta H_{30}$, $\Delta H_{90}$, $\Delta H_{180}$, along with a humidity equivalent of adaptive temperature (a proxy for a population's long term response to local humidity).

Therefore the full list of inputs is: the maximum temperature of the heatwave; the mean relative humidity during the heatwave; the mean windspeed during the heatwave; the mean BMI of the population; mean age; age gradient, or the slope of the all gender population pyramid; mean SDI; $\Delta T_{30}$, the temperature differential over the previous 30 days' average; $\Delta T_{90}$, the temperature differential over the previous 90 days' average; $\Delta T_{180}$, the temperature differential over the previous 90 days' average; the adaptive temperature, as described above; $\Delta H_{30}$, the humidity differential over the previous 30 days' average; $\Delta H_{90}$, the humidity differential over the previous 90 days' average; $\Delta H_{180}$, the humidity differential over the previous 180 days' average; and the adaptive humidity, as described above. The procedure for generating any derived variables is described in full in the 'Methods' section.

The machine learning method that we employ here is the Random Forest Classifier[39,40], an ensemble learning method that aggregates the output of a set of decision trees. Random Forests have been used extensively across a range of classification tasks in both environmental sciences[41–45] and healthcare[46–48] with a high level of success and have shown to be effective when handling imbalanced datasets[49]. Coupled to the Random Forest model is a Platt scaling logistic transformation, calibrating the output probabilities of the classification model such that they align with the true expected likelihood of an event based on observations[50]. This modification of the aggregated output of a random forest to a probabilistic output provides a degree of uncertainty about the prediction and provides a tunable probability threshold over which a decision maker can choose to act, effectively minimising either false negatives or false positives. In this case, we set a probability threshold of 0.6 to separate the lethal heatwaves from the non-lethal.

Our tuned model achieved accuracy of 0.991, precision of 0.690, recall of 0.660 and F1 score of 0.674 on the test set, which was a subset

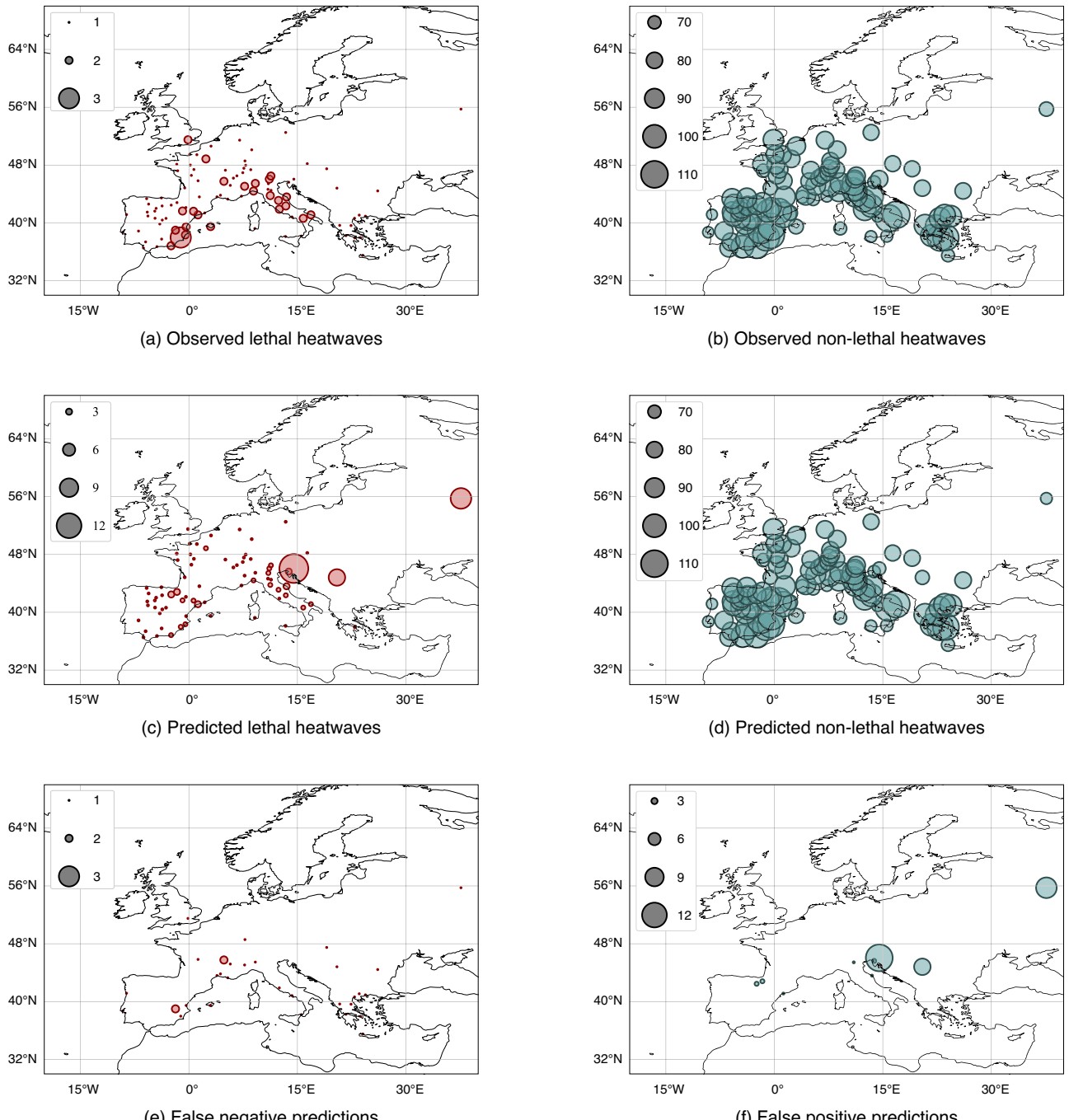

Fig. 2 | **European distribution of our classifier's lethality label predictions.** Number of test set **a** observed lethal heatwaves, **b** observed non-lethal heatwaves, **c** predicted lethal heatwaves, **d** predicted non-lethal heatwaves, **e** false negative predicted heatwaves (those incorrectly labelled as non-lethal), and **f** false positive predicted heatwaves (those incorrectly labelled as lethal) aggregated across cities in Europe, from 1980 to 2014, with predictions generated by our classifier. The scaling of the circles represents the number of heatwave events. Geographical underlays created using Cartopy[74].

containing 10% of all the recorded events from 1980 to 2014. Figures 2 and 3 show the distribution of lethal and nonlethal heatwaves in Europe and Asia, respectively, for the years 1980 to 2014 and the events that our model failed to predict correctly, the false negatives and the false positives.

## Sensitivity analysis

To analyse the importance of the input features, we test the model's robustness to feature permutation (randomly shuffling the input values of a given feature) and to feature dropout (the removal of that feature from the input space)[39,51], both of which are described in more detail in the 'Methods' section. The results, for the change in F1 scores for both of these approaches are shown in Fig. 4, such that the relative importance of each feature can be seen compared to the others.

Through both feature permutation and feature dropout, the Random Forest model is shown to be most sensitive to the variables we use to represent the health of a population alongside the maximum temperature. The most important temperature differential is the 180 day differential, $\Delta T_{180}$, followed by the 30 day temperature differential, $\Delta T_{30}$. The 90 day differential, $\Delta T_{90}$, is revealed to be somewhat less important, based on the increase to F1 score on permutation. Of particular note is the fact that the humidity is notably less important

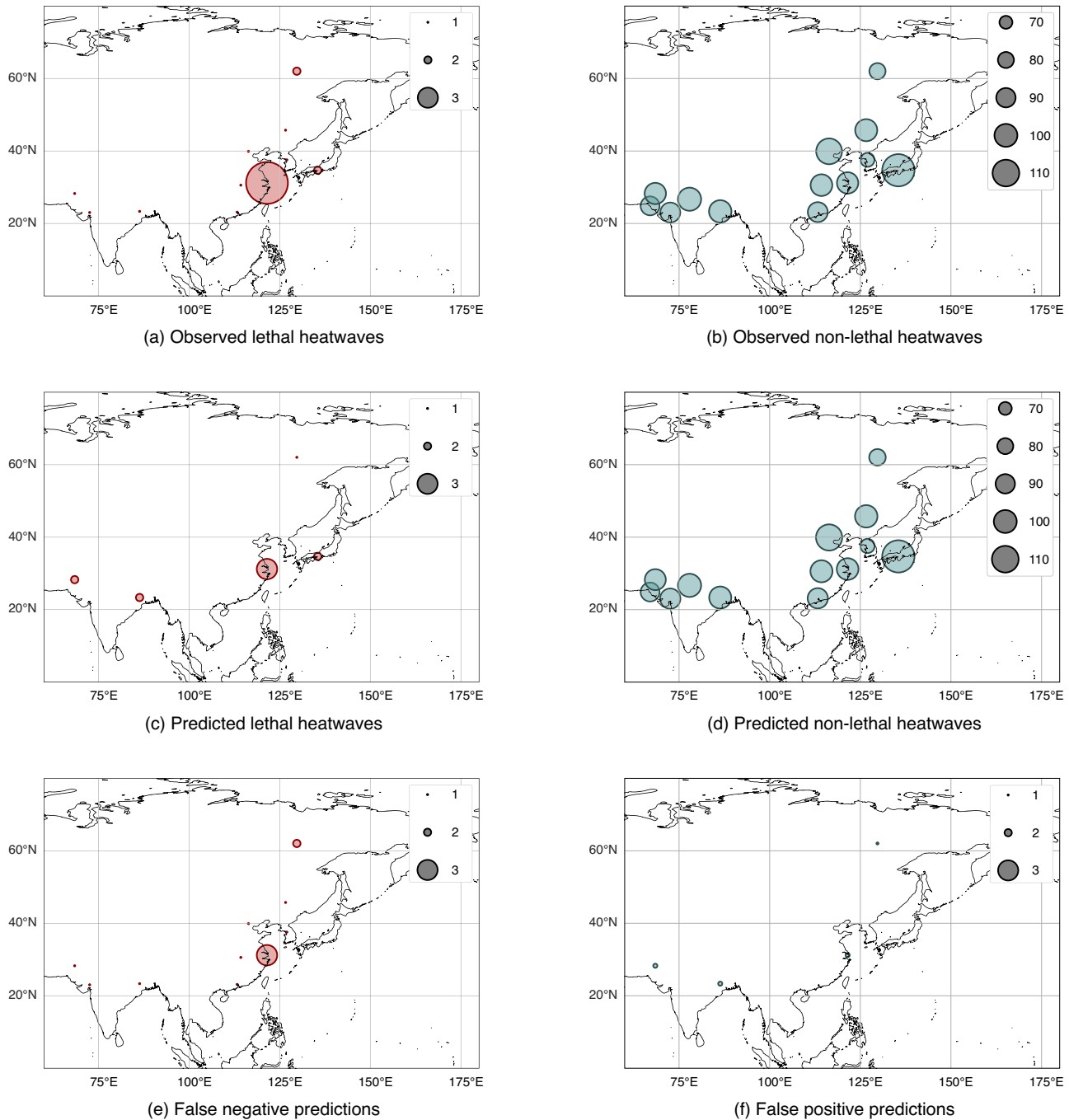

**Fig. 3 | Asian distribution of our classifier's lethality label predictions.** Number of test set **a** observed lethal heatwaves, **b** observed non-lethal heatwaves, (**c**) predicted lethal heatwaves, **d** predicted non-lethal heatwaves, **e** false negative predicted heatwaves (those incorrectly labelled as non-lethal) and **f** false positive predicted heatwaves (those incorrectly labelled as lethal) aggregated across cities in Asia, from 1980 to 2014, with predictions generated by our classifier. The scaling of the circles represents the number of heatwave events. Geographical underlays created using Cartopy[74].

and the variables representing the acclimation of a population to antecedent humidity conditions, the humidity differentials $\Delta H_{30}$, $\Delta H_{90}$, $\Delta H_{180}$ and adaptive humidity, are even less so with these two methods suggesting these variables are of little importance or possibly a hindrance. Whilst it should be noted that the sensitivity analysis results reflect the sensitivity of this model only, we might infer that the model is capable of capturing the failure of the most vulnerable members of a population to adapt to a rapid temperature change. This might be over a 6 month period, such as moving from a cold winter to a hot summer, or a sudden spike over the mean temperature that stresses those vulnerable.

The discrepancy in variable importance between the two methods is due to how each functions and potential correlation between variables. The extent to which information is lost on feature removal depends on correlation with other features[52]; for example, if high correlation exists between a pair of variables, then the information lost through removing one is minimal. On the other hand, with feature permutation, the importance is effectively shared between highly correlated variables. Consequently, the sensitivities from the feature dropout is less pronounced than those from the feature permutation. Therefore, if we consider the model robust to the removal of some features, we should also consider whether or not this is due to

correlation, rather than the variable being unimportant. This correlation is shown in Fig. 5.

The correlation matrix shows repeated information within the input space. First, the correlation between the temperature differentials is to be expected, given that they are all functions of the maximum temperature and have overlapping temporal dependencies. The 180 day differential encodes more of this temporal information and is likely representing the acclimatised temperature of a population, which might explain it being a more effective predictor. This also applies to the humidity differentials. The correlation between the temperature differentials and maximum temperature is limited, given the maximum temperature in the heatwave serves as a reference point and that they, along with the humidity differentials, do not leak the absolute conditions of a heatwave to the model. Of note is the correlation between mean age and mean BMI, which might run counter to intuition that longer lived populations are necessarily healthier. The negative correlation between the temperature and humidity variables is reflective of the skew towards temperate versus tropical regions within our dataset. The correlation between multiple variables explains the model's robustness in part but the decrease in our target metrics from removing even the variables that have less information in common with others is still small, which might be indicative of the strength of our approach overall.

Given the overlap between variables, we aim to determine a minimum subset of predictors that could result in an effective model. Therefore, we start with an empty set of predictors and build up the model one predictor at a time, selecting the predictor that best improves the model, in terms of F1 score, at each step. The results for this are shown in Table 1, where the F1 and precision scores increase as the recall decreases.

Although the first step adds maximum temperature, much of the model performance is gained by adding variables that provide information on the population health, such as Mean BMI, Mean SDI and the population age gradient, along with the deviation from the temperatures that populations have adapted to, effectively describing the physiological susceptibility of a population in response to a large swing in temperature when compared with expected conditions. This process does also reveal that the long term humidity differential, $\Delta H_{180}$, is detrimental to model performance.

To illustrate the weakness of using just temperature and humidity as predictors of heatwave mortality, we trained the algorithm on the same data and under the same experimental protocols as before. The maximal performance we obtained through this input set is also shown in Table 1 and underperforms our final model with all 15 variables by an order of magnitude. Demonstrating this further, and to an extent the unimportance of wet bulb temperature thresholds in many cases, we removed temperature and humidity from the input set and kept all other remaining variables. As discussed before, much of the information from these variables is not replicated through the temperature and humidity differentials and adaptive variables, resulting in a model that is relatively agnostic to the absolute conditions of a heatwave and yet still achieves almost the same performance as the complete variable set, as shown by the drop in F1 score of only $\approx 0.04$ in Table 1. Similar to Figs. 2 and 3, we also show the distribution of events incorrectly labelled, both as False Positives and False Negatives, by a Random Forest classifier using only temperature and humidity as predictors in Figs. 6 and 7.

## Discussion

Based on the sensitivity analysis that we have presented, we believe that a more appropriate classification of heatwaves lies in their impact mechanism. For the majority of the lethal heatwaves within our dataset, the wet bulb temperature was not excessively high and a less important factor than our inferred physiological state of the population and the temperature relative to that physiological state. This was

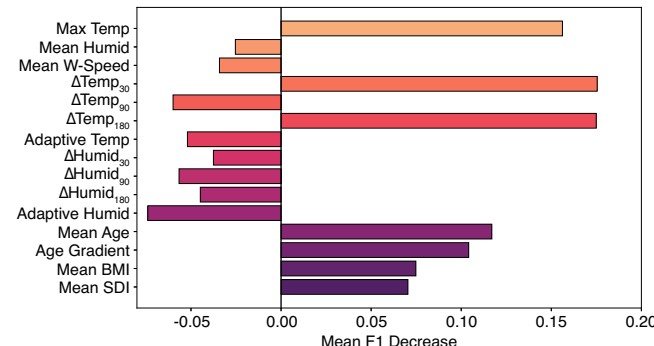

(a) Mean decrease in F1 from feature permutation

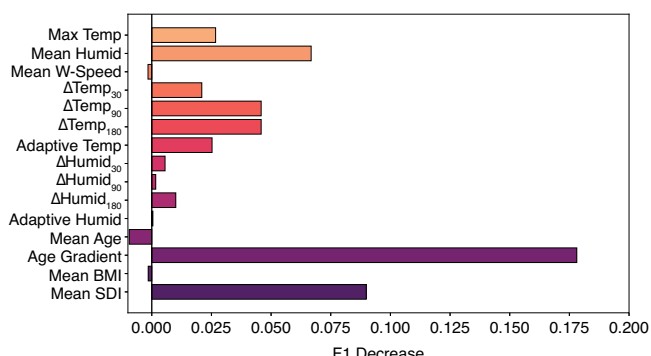

(b) Decrease in F1 from feature dropout

**Fig. 4 | Feature permutation and dropout quantifying feature impact on model performance.** Input feature impact on model performance measured using **a** the mean decrease in F1 score through feature permutation importance and **b** the decrease in F1 score through feature dropout for the Random Forest classifier, where: Max Temp is the maximum temperature, Mean Humid is the mean relative humidity, $\Delta\text{Temp}_{30}$ is the temperature differential over the previous 30 days' average, $\Delta\text{Temp}_{90}$ is the temperature differential over the previous 90 days' average, $\Delta\text{Temp}_{180}$ is the temperature differential over the previous 180 days' average, Adaptive Temp is the adaptive temperature, $\Delta\text{Humid}_{30}$ is the humidity differential over the previous 30 days' average, $\Delta\text{Humid}_{90}$ is the humidity differential over the previous 90 days' average, $\Delta\text{Humid}_{180}$ is the humidity differential over the previous 180 days' average, Adaptive Humid is the adaptive humidity, Mean Age is the average population age, Age Gradient is the slope of the population pyramid, Mean BMI is the average Body Mass Index across the whole population and Mean SDI is the average Socio-Demographic Index.

further underlined by the poor predictive capability of temperature and humidity alone; relying on wet-bulb temperature would fail to accurately predict the instance of heat-related deaths globally, an issue that could likely be exacerbated by climate change instability[53].

Thus, we prefer to define heatwaves as being either a *Shock Heatwave* or *Threshold Heatwave*, where the former refers to those with less extreme thermal conditions in absolute terms but aggressive thermo-temporal differentials that push members of a population, particularly the vulnerable, beyond their adaptive capacity and trigger lethal stress responses. The latter refers to those events that conform with the more conventional understanding of lethal heat, whether the combination of temperature and humidity lead to the human body being unable to dissipate heat and, subsequently, hyperthermia and related illnesses. We suggest that any lethal heatwaves occurring below a wet bulb temperature of 25 °C are almost certainly Shock Heatwaves and that they be termed as such.

The majority of the lethal events within our dataset fit the characteristics of a Shock Heatwave, given that our dataset is comprised primarily of heatwaves from temperate regions, with approximately 68.6% of the total from Europe. In temperate climates, lethality has

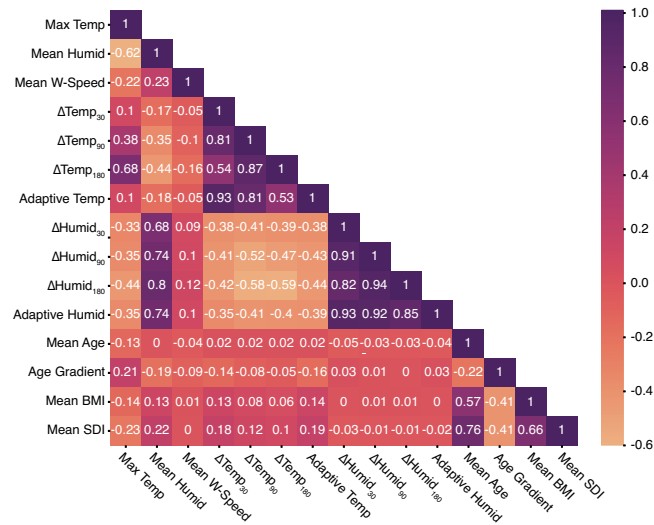

**Fig. 5 | Input feature correlation heat map.** Correlation between the predictors used in the Random Forest classifier, where: Max Temp is the maximum temperature, Mean Humid is the mean relative humidity, ΔTemp₃₀ is the temperature differential over the previous 30 days' average, ΔTemp₉₀ is the temperature differential over the previous 90 days' average, ΔTemp₁₈₀ is the temperature differential over the previous 180 days' average, Adaptive Temp is the adaptive temperature, ΔHumid₃₀ is the humidity differential over the previous 30 days' average, ΔHumid₉₀ is the humidity differential over the previous 90 days' average, ΔHumid₁₈₀ is the humidity differential over the previous 180 days' average, Adaptive Humid is the adaptive humidity, Mean Age is the average population age, Age Gradient is the slope of the population pyramid, Mean BMI is the average Body Mass Index across the whole population, and Mean SDI is the average Socio-Demographic Index.

**Table 1 | Performance on the test set by building up the set of predictors one at a time to include the predictor that best improves the model at each step, where: Max Temp is the maximum temperature, Mean Humid is the mean relative humidity, $\Delta T_{30}$ is the temperature differential over the previous 30 days' average, $\Delta T_{90}$ is the temperature differential over the previous 90 days' average, $\Delta T_{180}$ is the temperature differential over the previous 180 days' average, $\Delta H_{30}$ is the humidity differential over the previous 30 days' average, $\Delta H_{90}$ is the humidity differential over the previous 90 days' average, $\Delta H_{180}$ is the humidity differential over the previous 180 days' average, Age Gradient is the slope of the population pyramid, Mean BMI is the average Body Mass Index across the whole population and Mean SDI is the average Socio-Demographic Index**

| Input variables used | Input # | Metric | | | |
|---|---|---|---|---|---|
| | | **Accuracy** | **Precision** | **Recall** | **F1** |
| Max Temperature | 1 | 0.786 | 0.037 | 0.580 | 0.070 |
| ↳ & Mean BMI | 2 | 0.956 | 0.189 | 0.653 | 0.293 |
| ↳ & Adaptive Humidity | 3 | 0.982 | 0.403 | 0.625 | 0.490 |
| ↳ & Mean SDI | 4 | 0.984 | 0.458 | 0.676 | 0.546 |
| ↳ & $\Delta T_{180}$ | 5 | 0.988 | 0.562 | 0.619 | 0.589 |
| ↳ & Age Gradient | 6 | 0.988 | 0.559 | 0.648 | 0.600 |
| ↳ & $\Delta H_{90}$ | 7 | 0.990 | 0.625 | 0.653 | 0.639 |
| ↳ & $\Delta T_{30}$ | 8 | 0.990 | 0.640 | 0.676 | 0.657 |
| ↳ & Mean Age | 9 | 0.990 | 0.623 | 0.676 | 0.649 |
| ↳ & Mean Humidity | 10 | 0.990 | 0.647 | 0.676 | 0.661 |
| ↳ & $\Delta H_{30}$ | 11 | 0.991 | 0.682 | 0.659 | 0.671 |
| ↳ & $\Delta T_{90}$ | 12 | 0.991 | 0.700 | 0.676 | 0.688 |
| ↳ & Mean Windspeed | 13 | 0.992 | 0.740 | 0.648 | 0.690 |
| ↳ & Adaptive Temperature | 14 | 0.992 | 0.720 | 0.660 | 0.689 |
| ↳ & $\Delta H_{180}$ | 15 | 0.991 | 0.690 | 0.660 | 0.674 |
| Max Temperature & Mean Humidity Only | 2 | 0.867 | 0.033 | 0.307 | 0.060 |
| Without Max Temperature & Mean Humidity | 13 | 0.990 | 0.644 | 0.647 | 0.646 |

been less driven by absolute meteorological extremes. These populations have a general 'U'-shaped relationship between temperature and mortality[54] with mortality tending to rise with increasingly hot or cold temperatures from an optimum value. For example, in Britain, mortality is lowest with a mean daily temperature of 17–18 °C[55], although there is evidence that there is increasing tolerance to higher temperatures within this population as the baseline temperature shifts upwards[22]. Heat-related illness[11] can involve a wide range of pathologies but deaths are still mainly cardiovascular with meta-analysis identifying all-cause cardiovascular illness as the primary cause of death[56]. Our empirical approach and flexible classification of heatwave lethality enables the outcome of these events to be accurately predicted and, therefore, has more global utility than wet bulb temperature based classifiers.

The geographical imbalance in available data, however, is problematic and likely driven by a combination of low autopsy rates relative to the population[57] and potentially dismissive postmortem misdiagnosis[58], labelling cause of death as natural or similar. As a consequence, there is no representation from Africa, South America and parts of Oceania in the data we used, which biases the classifier geographically towards higher and middle income countries that are more studied[10]. Whilst this does limit the ability to prove that our model is capable of generalising across all climates, the heatwaves in the higher temperature subregions that are particularly underrepresented are likely to be Threshold Heatwaves, and could be captured by the meteorological decision branches of the model. The fact that these regions typically have lower incomes does raise another issue: due to the link between income and both the prevalence of air conditioning[59] and the level of healthcare provision[60], utilising appropriate measures for economic development could act as suitable proxy for access to technologies that mitigate dangerous heat. At a more granular level, urban heat island effect and the associated

negative health consequences[61] might be approximated by incorporating urban density or urban vegetative cover. As a result, our approach might be utilised with these additional features to create spatially distributed maps of lethal heat risk within a country.

A related concern is the verification of heatwave lethality. Due to the limited pre- and postmortem diagnosis conducted in the vast majority of countries[57,58], including many in this study, there could be errors in the cause of death data, either through misattribution or it being missing altogether. The noise to signal ratio is also highly likely to be non zero for displaced and excess mortality, which becomes problematic for lower fatality events when total fatality rates are likely within typical noise levels. So even with a dataset compiled from peer-reviewed research, there could be errors within that data. Verifying the cause of death and attribution would also be challenging, if not impossible, as there is no way or revisiting individual cases or interrogating the stated cause of death when there is almost certainly no body or remaining human tissue of suitable integrity for pathology. This is an issue that is also borne out in the inconsistent reporting on the number of deaths across the data we used and one that is also not improving, due to falling rates of post mortem diagnosis[62].

As a result, our work is impacted in two ways: the first being that this research is limited to the determination of lethality and that this determination does not provide nuance on the level of impact of these heatwaves; the second is that there is a chance that some heatwaves

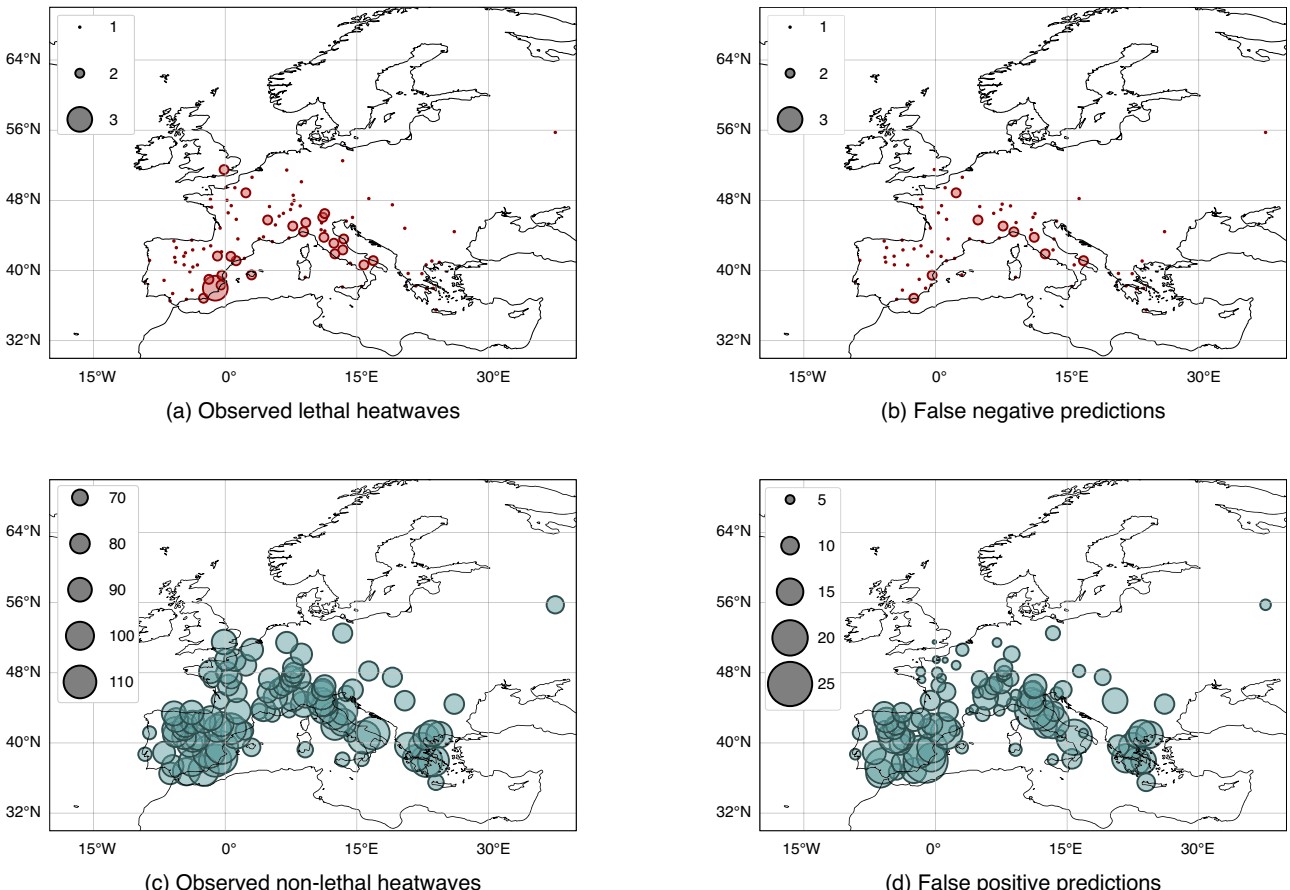

(a) Observed lethal heatwaves

(b) False negative predictions

(c) Observed non-lethal heatwaves

(d) False positive predictions

**Fig. 6 | European distribution of a wet bulb temperature classifier's lethality label predictions.** Number of test set **a** observed lethal heatwaves, **b** false negative predicted heatwaves (those incorrectly labelled as non-lethal), **c** observed non-lethal heatwaves and **d** false positive predicted heatwaves (those incorrectly labelled as lethal) aggregated across cities in Europe, from 1980 to 2014, with predictions generated by a temperature-humidity only classifier. The scaling of the circles represents the number of heatwave events. Geographical underlays created using Cartopy[74].

have been incorrectly labelled and thus the misclassification of heatwaves remains a possibility. The probability likely varies in accordance with the rate and accuracy of pre- and postmortem diagnosis, and subsequent analysis, and is also far more likely to apply to those heatwaves with few fatalities or those incorrectly reported as having caused none at all. A consequence of this is that the model might learn to predict events as nonlethal that were in fact lethal. Misclassification at prediction, based on other forms of error such as structural uncertainty, could be problematic in the same way, given that underprediction of lethal heatwaves is the more damaging scenario in terms of human impact. One, therefore, might favour maximising recall over precision and potentially affecting F1 score. A decision maker, or other practitioner, using this framework might adopt such a mindset to minimise that impact, particularly in targeting a mitigation response for the vulnerable. This could be achieved by choosing a more conservative probability threshold applied to the probabilities obtained through Platt Scaling. We note that, due to the class imbalance, decreasing the number of false negatives could create a disproportionately high number of false positives, representing too high an absolute number of events upon which decision makers need to act but this balance is not something we address here. Overall, this framework does provide a suitable basis for decision makers to create or update operational warning systems based on local spatiotemporal conditions; for example, the UK Health Security Agency's absolute threshold based system[27] could be modified using our approach to a seasonally-adjusted risk metric that is directed to vulnerable subpopulations.

Finally, the model architectures we explored are relatively well established and were well suited to the problem of imbalanced classification; using alternatives, such as Artificial Neural Networks or Gaussian Processes, or variants of Random Forests, such as Gradient Boosted Decision Trees[63], all showed similar patterns in the results without improvement and we therefore did not include the additional results. As a result, we believe that if there were improvement that could be obtained from even further modification of the model architecture that it would likely be small when compared with the benefits of further improving the data fed in to the analysis.

To summarise, we have presented a holistic model for predicting whether or not a heatwave will be lethal. By distilling our understanding of physiology, sociology and meteorology into a proxy based feature space, we have developed a model that achieves high performance, that outperforms the use of temperature and humidity alone, and that can achieve similar performance in the absence of using wet bulb temperature entirely. Due to the order of magnitude improvement on wet bulb temperature classification, we believe that our framework is suitable for adoption by those working in early warning capacities to provide more specific alerts to vulnerable populations and subpopulations. Furthermore, our findings, in that there are heatwaves with lethal impact at relatively low temperature and humidity levels, suggest two distinct mechanisms for heatwave lethality that need to be recognised at a public health level: one related to the body's inability to dissipate heat, and the other related to a rapid environmental change triggering a stress response, both of which can prove fatal.

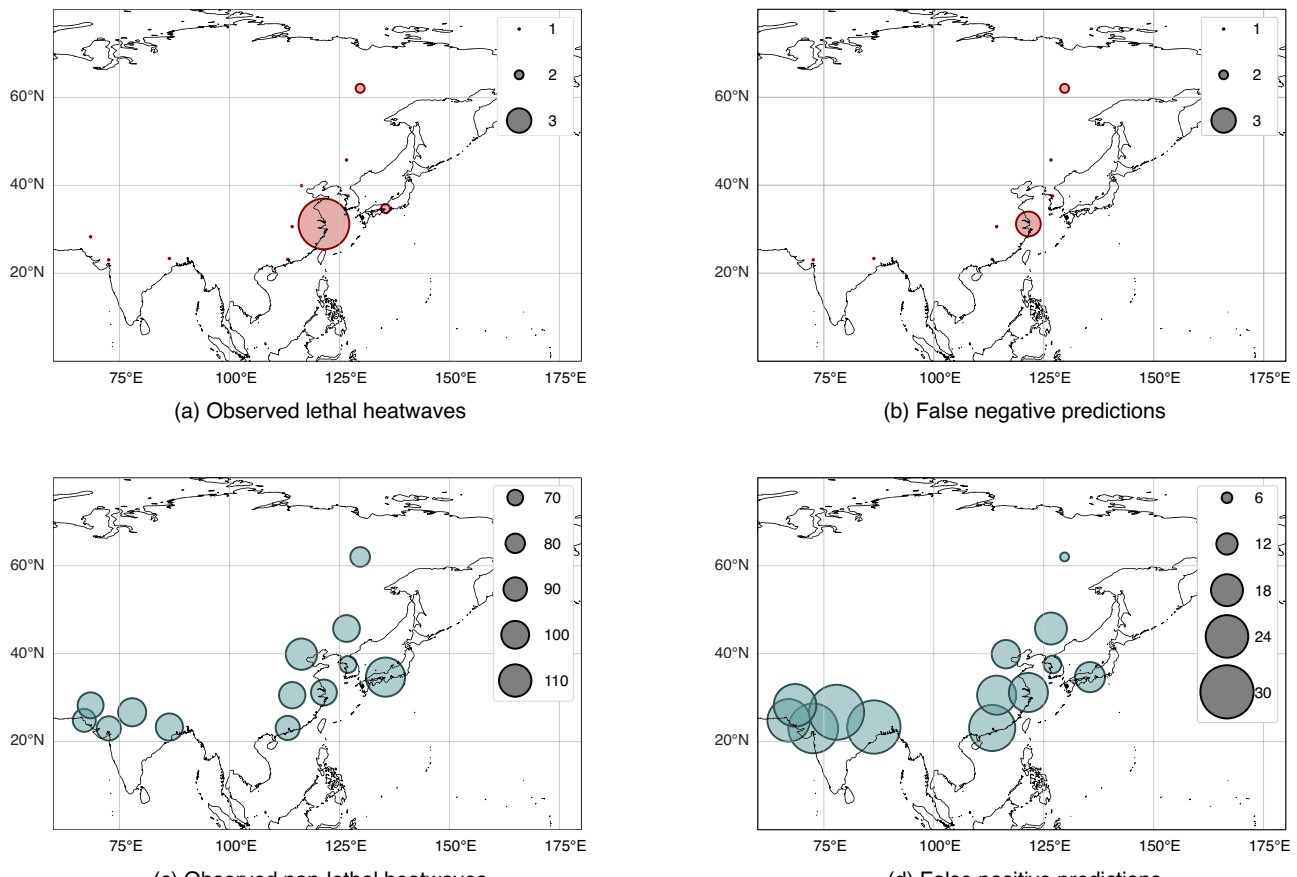

Fig. 7 | **Asian distribution of a wet bulb temperature classifier's lethality label predictions.** Number of test set **a** observed lethal heatwaves, **b** false negative predicted heatwaves (those incorrectly labelled as non-lethal), **c** observed non-lethal heatwaves and **d** false positive predicted heatwaves (those incorrectly labelled as lethal) aggregated across cities in Asia, from 1980 to 2014, with predictions generated by a temperature-humidity only classifier. The scaling of the circles represents the number of heatwave events. Geographical underlays created using Cartopy[74].

The strong connection between adverse outcomes, climate trends and public health vulnerabilities is particularly alarming, given the continuing deterioration in both climate and population health[32]; without intervention, the size of the vulnerable population and number of lethal Shock Heatwaves will likely increase even in temperate regions alongside potential increases in Threshold Heatwaves. Our intention is to apply the model to future climate scenarios and generated meteorological time series, along with projections for high-level sociodemographic and economic data and the inclusion of descriptors of local environment. By doing so, we aim to develop a deeper understanding of which regions and populations are most vulnerable to both Shock and Threshold Heatwaves, particularly under high-warming scenarios and how policy can be designed to mitigate and adapt to the spectrum of risk factors presented here.

## Methods
### Data preprocessing
The meteorological data we use is taken from The European Centre for Medium-Range Weather Forecast's (ECMWF) fifth generation of global climate reanalysis data, ERA5. Although a reanalysis product, we note that is generally considered accurate when compared to observations[64,65]. From this gridded data product, the variables that we are interested in are the maximum temperature during the heatwave, the humidity and windspeed. For the latter two, the mean is taken over the heatwave period. All three meteorological variables have been taken at a pressure level of 1000 hPa, being close to the surface.

The temperature differential, $\Delta T_n$, is expressed in terms of the heatwave maximum temperature and the mean of the maximum temperature over the $n$ days preceding the heatwave in Equation (1); a similar expression for the humidity differential, $\Delta T_n$, is in Equation (2) but in terms of the heatwave mean humidity and the mean of the humidity over the $n$ preceding days.

$$\Delta T_n = T_{max} - \frac{1}{n}\sum_{i=1}^{n} T_{max\,i} \tag{1}$$

$$\Delta H_n = H_\mu - \frac{1}{n}\sum_{i=1}^{n} H_{\mu\,i} \tag{2}$$

The adaptive temperature is taken as the mean of the maximum temperature over the $n$ days preceding the date of the heatwave over that year and the preceding $m-1$ years, as expressed in Equation (3). Again, a similar approach is adopted for the adaptive humidity, as expressed in Equation (4), but in terms of the mean. In this paper, $m = 10$. Thus, for example, if the heatwave occurred at the end of January, the average of the maximum temperatures for that January and the previous 9 Januaries would be taken.

$$\text{Adaptive T} = \frac{1}{m \cdot n}\sum_{j=1}^{m}\sum_{i=1}^{n} T_{max\,ij} \tag{3}$$

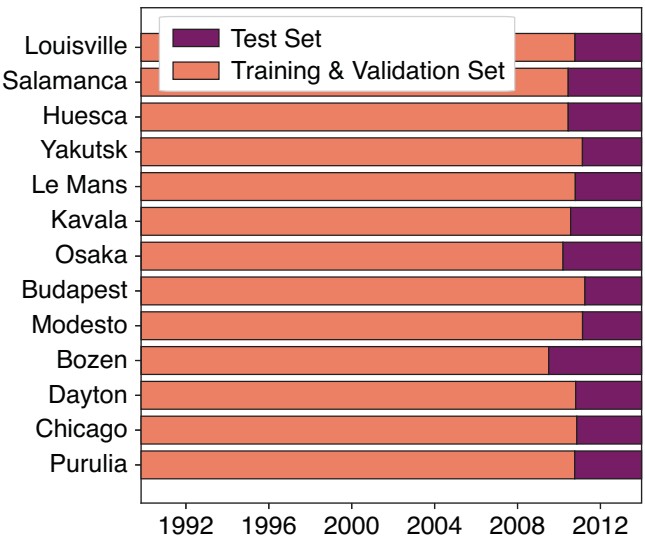

**Fig. 8 | Chronological training, validation and test set splits for a subset of locations.** Date at which the data is separated into the training and validation sets and the test set to minimise temporal leakage for a subset of the regions included in this study. The different sets contain the same amount of data by volume, with 90% of the data in the training and validation sets combined and 10% in the test set.

$$\text{Adaptive H} = \frac{1}{m \cdot n} \sum_{j=1}^{m} \sum_{i=1}^{n} H_{\mu ij} \qquad (4)$$

The population age gradient is calculated using the linear regression formula in Equation (5), where the $y$ values correspond to the age bracket, discretised over 5 year intervals and the $x$ values correspond to the proportion of the population within each age bracket.

$$m = \frac{\overline{x} \cdot \overline{y} - \overline{x \cdot y}}{\overline{x} \cdot \overline{x} - \overline{x \cdot x}} \qquad (5)$$

### Assessment

Accuracy score, expressed in terms of correctly labelled events, either true positives, TP, or true negatives, TN, out of the whole set, including false positives, FP and false negatives, FN, is a common metric but not entirely useful when dealing with class imbalance. As the number of lethal events only represents 0.78% of the total set of events; consequently, one could predict that all events were nonlethal and end up with an accuracy score of 0.992, close to the maximum possible of 1. Therefore, we place less importance on accuracy and focus on metrics that are better suited to assessing the capture of true positives and the false negatives and positives in the face of class imbalance. Specifically, we use precision, as the proportion of correctly predicted positives out of all predicted positives, as expressed in Equation (6); recall, as the proportion of correctly predicted positives out of all actual positives, as expressed in Equation (7); and F1 score, the harmonic mean of precision and recall, as expressed in Equation (8).

$$Precision = \frac{TP}{TP + FP} \qquad (6)$$

$$Recall = \frac{TP}{TP + FN} \qquad (7)$$

$$F1 = \frac{2}{\frac{1}{Pr} + \frac{1}{Rc}} \qquad (8)$$

With these metrics, we essentially capture the false positive and the perhaps more crucial false negative rates of any algorithmic approach, circumventing issues of assessment for imbalanced datasets. For each of these metrics, the optimal score is 1.

### Data sampling

For all experiments, we utilised the same training, validation and test set splits: to help mitigate temporal leakage, whilst still preserving the geographic balance of lethal events in the training set to ensure effective learning of these rare, extreme occurrences[66], we have taken the most recent 10% of events by region as the test set. The dates around which that 90–10% split occurs is shown for a subset of cities in Fig. 8. We selected a further 10% of events from the training set without stratification to use as the validation set for model optimisation, thereby creating a split of 80%, 10% and 10% across the training, validation, and test sets, respectively.

The class imbalance issue effects the training of machine learning algorithms, not just their assessment[67–69]. Therefore, in spite of our prior commentary on Random Forests performing well with imbalanced data, we still attempt to redress the class imbalance during training. In order to do so, we can reconstruct the training set to have more balance, through upsampling from the minority class or downsampling from the majority class. Random downsampling from the majority class limits the size of the training set, the effects of which, as per further results in the appendix, minimise the diversity of nonlethal heatwaves leading to an overprediction of lethal heatwaves, creating high recall but low precision and F1 score overall when more than 75% of the nonlethal events are removed. Upsampling achieves class balance and preserves the number of negative examples from which to learn but, if we are going to increase the number of positive examples within the training set, then we might prefer to augment the dataset instead. Using a technique such as the Synthetic Minority Oversampling Technique[70] enables the synthesis of new examples from the minority class, using the principle of $k$ nearest neighbours from any given example within the feature space, and helps achieve higher F1 scores overall on the validation set than any of the downsampling strategies. Further information can be found in the Supplementary Information.

### Sensitivity analysis

For the 10 variables that we included in the final model, we estimate our model's sensitivity and dependence on each input variable through: the feature permutation importance algorithm[51], which, for some error function $\mathcal{L} = (y, f(\mathbf{x}))$, applies a permutation to the entries representing a given input feature, $j$ in the matrix of all inputs, $\mathbf{X}$, to derive a new error, $\mathcal{L}_j$, and then compares $\mathcal{L}$ against $\mathcal{L}_j$; and through feature dropout, whereby each feature is in turn removed and the model retrained with the corresponding decrease in model metrics recorded.

### Reporting summary

Further information on research design is available in the Nature Portfolio Reporting Summary linked to this article.

## Data availability

The lethal/non-lethal heatwave labels have been made available by Dousset and contributors[71] through the following repository https://doi.org/10.5281/zenodo.18528487. ERA5 reanalysis variables can be obtained from https://cds.climate.copernicus.eu/cdsapp#!/dataset/

reanalysis-era5-pressure-levels. Population data is available from https://population.un.org/wpp/. BMI data is available from 10.5281/zenodo.10534960. SDI data is available from 10.6069/dwqg-3z75.

## Code availability

The code for this paper is available at https://doi.org/10.5281/zenodo.18430720. This codebase provides a blueprint to download the requisite data through querying their respective APIs, compile the database, train and test the model, and generate key figures. The random forests were implemented using Scikit-learn[72] and SMOTE using imbalanced-learn[73].

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

## Acknowledgements

The authors would like to thank Rick Lewis, of the Centre for Landscape Regeneration at Cambridge University, for reviewing this work. RER's acknowledges funding from The Royal Commission for the Exhibition of 1851 and EPSRC grant EP/R512461/1. RD acknowledges support from the Bill & Melinda Gates Foundation [OPP1144] and the Cambridge Humanities Research Grant. W.T. acknowledges funding from Deepmind, Huawei, and EPSRC grant EP/W002965/1.

## Author contributions

C.M. and B.D. compiled the initial lethal heatwave dataset. R.E.R., D.A.R., J.S.H., A.M. and E.S. conceived the work. R.E.R. and D.A.R. developed the approach. R.E.R. implemented and trained the model with guidance from W.T. and A.M. and R.E.R. wrote the code. R.E.R. and W.T. conducted the analysis with input and feedback from all authors. R.E.R. with input from E.S. completed the visualisation. R.E.R., D.A.R., R.D. and W.T. wrote the draft manuscript contributions and feedback from all authors. J.S.H., A.M. and E.S. provided supervision and funding acquisition.

## Competing interests

The authors declare no competing interests.
