## [Transparent Peer Review file · Nature Communications]

Reclassifying Lethal Heat

Corresponding Author: Dr Robert Rouse

Version 0:

Reviewer comments:

Reviewer #1

(Remarks to the Author)

Key Results:

The paper presents a new classification framework for heatwaves by utilizing a Random Forest classifier, proposing two categories: "Shock Heatwaves" (triggered by rapid thermo-temporal changes) and "Threshold Heatwaves" (exceeding physiological limits). By analyzing data from 125,411 heatwave events across 140 cities globally, the authors demonstrate that lethal heatwaves can occur at temperatures below traditional wet bulb temperature thresholds, and thus the proposed classification, grounded in meteorological, physiological, and socio-demographic data, provides a more accurate and nuanced prediction of lethal heatwaves. A key finding is that lethal heatwaves can occur even at lower temperature and humidity levels than from simplistic temperature-humidity models, suggesting that rapid environmental changes and population vulnerability play critical roles in heatwave lethality. The proposed nomenclature on Shock and Threshold Heatwaves appears rather well-suited in this context and addresses valid concerns that merely climatic criteria that are often used in climate literature overlook this important fact.

Validity:

The methodology is technically sound, but certain aspects of the approach raise concerns about its practical application and scientific contribution that should be clarified. The use of a Random Forest classifier offers strong predictive accuracy, but the decision to focus on binary classification (lethal vs. non-lethal) may be too simplistic given the complexity of heatwave impacts. The authors should motivate why predicting severity of heatwaves (e.g., mortality count or intensity) is not the path taken, since it appears that severity could provide more actionable insights for public health interventions. Furthermore, the concrete lethality definition is not really presented which makes it difficult to comprehend what criteria were used besides the cited literature. The article should be self-contained in this way and present the lethality criterion at least in methodology section.

Furthermore, the authors rely on randomly splitting the data by years for training and testing, which may introduce temporal leakage and does not reflect how the model would be applied to predict future heatwave events. Heatwave data has temporal dependencies, and using a chronological split (training on earlier years, testing on future years) would provide a more realistic and robust evaluation of model performance.

Significance:

The introduction of the two distinct mechanisms for heatwave lethality (1) the body's inability to dissipate heat, and (2) stress responses triggered by rapid environmental changes - represents an important conceptual advancement. However, the practical significance of the new classification system is limited by the interpretability of the Random Forest model. Given the critical need for transparency and ease of use in public health policy, the black-box nature of Random Forest, while superior to deep neural network, may still hinder the model's adoption in real-world scenarios. It is true that some explainability analysis is performed, which is quite welcome, such as feature permutation and feature dropout. Yet, for practitioners it would be useful to have an approximate expression that takes the considered factors into account.

While the model improves accuracy over traditional metrics, introduces a machine learning model able to assign a new category for heatwaves (shock heatwaves) in the current state, it does not appear to provide enough novel insight or

practical value to justify publication in a high-impact journal like Nature Communications. The contribution, while conceptually interesting, could be strengthened by focusing on the model's real-world applicability and ease of use. The possible applications are discussed briefly in the discussion section, but it would help to provide more detailed study of this question. I would like to encourage the authors because the idea of improving the way we describe and quantify heatwave impact is indeed a very pertinent one.

Suggested Improvements:

1. Binary classification: Provide clear justification for using binary classification
2. Lethality criterion: Provide clear definition of criterion used to provide lethality label, since the task is supervised learning.
3. Chronological Data Splitting: The current method of randomly splitting the data by years introduces temporal leakage. A chronological train-test split, or a rolling window validation would offer a more realistic evaluation of the model's ability to predict future lethal heatwave events, especially considering climate change effects.
4. Interpretability: The reliance on a Random Forest classifier, limits the utility of the results. The authors should provide clear, actionable insights from the model's predictions and more clear directives on how this model can be used by the practitioners.

Clarity and Context:

The manuscript is generally well-written, but certain passages are a bit awkward especially in the discussion section.

Areas for Improvement:

Discussion section, page 14:

1. "Achieves high levels of accuracy":

- The statement on line 322 refers to accuracy even though technically we know that accuracy is not a good measure of how well the algorithm performs for imbalanced classification as stated by the authors elsewhere.
- Suggestion: Consider using terms precision and recall rather than accuracy. A short explanation of what these terms mean can be included.

2. "Two different mechanisms of impact":

- The phrasing "we believe that there are, in effect, two different mechanisms of impact" is clear, but the explanation following it could benefit from a slight restructuring for clarity. The two mechanisms - (1) exceeding the body's ability to dissipate heat, and (2) a stress response to rapid environmental changes - are important points, but they could be laid out more succinctly.
- Suggestion: Rephrase as "Our findings suggest two distinct mechanisms for heatwave lethality: one related to the body's inability to dissipate heat, and the other related to a rapid environmental change triggering a stress response, both of which can prove fatal." This version is more direct and cohesive.

3. "Links between climate and public health":

- The sentence "The strong links between negative outcomes and both climate and public health ought to be of concern" is a bit vague and abstract. It could benefit from more specificity to tie it back to the findings.
- Suggestion: Instead, you could say: "The strong connection between adverse outcomes, climate trends, and public health vulnerabilities is particularly alarming, given the continuing deterioration in both climate and population health." This strengthens the point and ties it to the context of climate change.

4. "Force it using future scenarios":

- The phrase "force it using future climate scenarios and generated meteorological time series" is somewhat awkward. The term "force" might not be the best word choice in this context, where "apply" or "extend" might work better.
- Suggestion: Rephrase as "Our intention is to apply the model to future climate scenarios and generated meteorological time series, along with projections for high-level sociodemographic data." This sounds smoother and retains the intended meaning.

5. Final Impact Statement:

- The conclusion about applying the model to future warming scenarios is strong but could be enhanced by tightening the language for clarity.
- Suggestion: "By doing so, we aim to develop a deeper understanding of which regions and populations are most vulnerable to both Shock and Threshold Heatwaves, particularly under high-warming scenarios." This version is more concise and maintains the same meaning.

Methods section

6. Consider rephrasing "fifth generation of global climate reanalysis data, ERA5, and note that is generally considered accurate when"
7. Line 363: fix: "out of out of the whole set"
8. Section 4.3: it would help to summarise what was the effect of resampling, using the contents of the appendix.

Data Availability

9. There seems to be missing link and references in the sentence on lines 407-408.

Code Availability

10. Since in the end there is no equation to describe the impact of heatwaves but a random forest model it is very important to facilitate the use of this model and clarity of the code by providing concrete explanations in the repository, so consider improving the repository along these lines.

Appendix

11. Line 491: "though up to a point" sounds informal, consider rewording. It would help to show that re-sampling does not qualitatively change the feature distributions.

References:

The manuscript appropriately references prior literature but could include more discussion of alternative machine learning models and their applications in similar contexts. A more in-depth comparison of this approach with existing methods for predicting heatwave lethality would provide better context for the contribution. Examples of relevant publications worth discussing:

Pir Mohammad, Qihao Weng, Nexus Volume 1, Issue 3, 17 September 2024, 100027

Expertise and Scope:

I am confident in my evaluation of the machine learning and meteorological aspects of the manuscript, though I would recommend further review by experts in public health and socio-demographics to ensure these dimensions are sufficiently addressed.

(Remarks on code availability)

Reviewer #2

(Remarks to the Author)

This manuscript introduces a nice research attempt to reclassify lethal heatwaves. As heatwaves are becoming more frequent and intense worldwide, the topic is worth exploring. It would be great if the authors could please shed some light on my questions raised through reading the manuscript:

Major comments:

I was struggling to follow the Results section. Could the authors please present the key findings in a more direct and succinct way, e.g., what are the main findings;

In the Introduction section, the authors pointed out the struggles that the climate and epidemiology research communities have in defining heatwaves. Please could they clarify why their approach is superior to the approaches used by the climate and epidemiology communities?

The authors used the 90th percentile of temperature distribution as the cut-off to define heatwaves. I would have thought this is similar to the approach used by the epidemiology community. Please could the authors shed some light on this;

Lines 118-119: In my naïve opinion, one of the main goals of classifying/defining heatwaves is to aid in the development of heatwave response strategies to protect public health. Please could the authors clarify why classifying heatwaves if not to predict their impacts;

Minor comments:

I am not too sure if it is the most recommended practice to cite references in the Results section;

Please could the authors explain what each variable means in the figure legends (e.g., Figure 5).

(Remarks on code availability)

Reviewer #3

(Remarks to the Author)

Summary of paper and general comments:

This manuscript proposes and applies a classification algorithm based on machine learning methods to identify lethal heatwaves and applies it to a collection of heatwaves identified in previous work. The manuscript is well-written and focuses on an important topic. However, the motivation and application of this classification algorithm is not clear, and some background information is missing. Specific comments are included below:

Major comments:

- The utility of this classification algorithm and relevance to policy or public health is not clear. Along the same line, using a binary lethal/non-lethal classification without considering the impact such as the number of deaths is not well justified.
- The authors state that “Epidemiological studies focus mostly on total population mortality and morbidity outcomes without differentiating between socially vulnerable groups.” This is overlooking an abundance of research evaluating social vulnerability to extreme heat from epidemiology. A few relevant articles include:
 - o Son, J. Y., Liu, J. C., & Bell, M. L. (2019). Temperature-related mortality: a systematic review and investigation of effect modifiers. *Environmental Research Letters*, 14(7), 073004.
 - o Benmarhnia, T., Deguen, S., Kaufman, J. S., & Smargiassi, A. (2015). Vulnerability to heat-related mortality: A systematic review, meta-analysis, and meta-regression analysis. *Epidemiology*, 26(6), 781-793.
 - o Arsad, F. S., Hod, R., Ahmad, N., Ismail, R., Mohamed, N., Baharom, M., ... & Tangang, F. (2022). The impact of heatwaves on mortality and morbidity and the associated vulnerability factors: a systematic review. *International Journal of Environmental Research and Public Health*, 19(23), 16356.
 - o Bakhtsiyarava, M., Schinasi, L. H., Sánchez, B. N., Dronova, I., Kephart, J. L., Ju, Y., ... & Rodríguez, D. A. (2023). Modification of temperature-related human mortality by area-level socioeconomic and demographic characteristics in Latin American cities. *Social Science & Medicine*, 317, 115526.
- The manuscript uses data from previous work (Mora et al., 2017), and more substantial description of the published manuscript and what this new study adds would be important. The added value of this particular classification tool should be clarified.
- The manuscript states: “Consequently, heat-related mortality data are sparse and have not been analyzed consistently [6, 7, 9].” The citations listed are outdated, and there is extensive literature on this topic that has not been cited. Here are a few:
 - o Ebi, K. L., Capon, A., Berry, P., Broderick, C., de Dear, R., Havenith, G., ... & Jay, O. (2021). Hot weather and heat extremes: health risks. *The Lancet*, 398(10301), 698-708.
 - o Khatana, S. A. M., Werner, R. M., & Groeneveld, P. W. (2022). Association of extreme heat with all-cause mortality in the contiguous US, 2008-2017. *JAMA Network Open*, 5(5), e2212957-e2212957.
 - o Kephart, J. L., Sánchez, B. N., Moore, J., Schinasi, L. H., Bakhtsiyarava, M., Ju, Y., ... & Rodríguez, D. A. (2022). City-level impact of extreme temperatures and mortality in Latin America. *Nature medicine*, 28(8), 1700-1705.
 - o Rocklöv, J., Ebi, K., & Forsberg, B. (2011). Mortality related to temperature and persistent extreme temperatures—a study of cause-specific and age stratified mortality. *Epidemiology*, 22(1), S13.
- A more detailed description of how the heatwaves were labelled as lethal or nonlethal should be included. Is there a possibility of misclassification in this categorization? How can we be sure that the heatwaves categorized as non-lethal heatwaves did not cause any deaths and is there a way to validate this?
- Physiological vulnerability to extreme heat encompasses a range of health factors including pre-existing health conditions and socio-demographics. Relying on BMI as an indicator of population health in relation to heat is unlikely to fully capture this vulnerability. Additionally, BMI is a flawed indicator of health (see Ahima, R. S., & Lazar, M. A. (2013). The health risk of obesity—better metrics imperative. *Science*, 341(6148), 856-858.) The manuscript states that “comprehensive data on underlying comorbidities and physiological adaptation is, for somewhat obvious reasons of practicality, impossible to collect for entire populations within all global subdomains (whether that subdomain is taken to be at the city, county, state, or national level) and is likely to remain so for some time”, but there are many potential data sources that could be used that will have information beyond age and BMI. Some examples include:
 - o Global Burden of Disease study: <https://ghdx.healthdata.org/series/global-burden-disease-gbd>
 - o Global socio-demographic index (SDI): <https://ghdx.healthdata.org/record/global-burden-disease-study-2021-gbd-2021-socio-demographic-index-sdi-1950%E2%80%932021>
 - o GLOPOP-S: <https://www.nature.com/articles/s41597-024-03864-2>
- As there are some false negative and false positive predictions, a discussion of the potential consequences in the misclassification of the lethality of heatwaves would be important.
- The discussion of what data is being used to train this model and what regions are missing from the data used would be important to contextualize the results.

Minor comments:

- Abstract: “Furthermore, we found that the majority of level heatwaves within our dataset..” - is this a typo?
- “Although the severity of these events varies in terms of the total mortality, such as 4,867 excess deaths in Paris arising from the 2003 heatwave [17] and 11 for the 1999 heatwave that affected Minneapolis [18]...” This does not reference a study conducted in Minneapolis.

(Remarks on code availability)

The GitHub and README file have several placeholders and portions that are not finalized that should be edited.

Version 1:

Reviewer comments:

Reviewer #1

(Remarks to the Author)

The revised manuscript proposed a new classification system for lethal heatwaves based on socio-demographic and meteorological variables rather than traditional wet bulb thresholds. The authors introduce a Random Forest classifier trained on historical heatwave mortality data, aiming to better capture the complex interplay of vulnerability and environmental stressors.

The manuscript has improved, addressing many of the comments of the initial review. However, there are still points that

need to be addressed. Below, I provide detailed comments on the key issues raised in the first round that remain relevant.

- 1.) Binary classification justification – addressed satisfactorily. The authors have expanded their rationale for using binary classification, focusing on the inadequacy of wet bulb temperature thresholds.
- 2.) Definition of Lethality Labels – Partially Addressed. While the authors have provided a more expansive description of lethality labeling, it remains difficult to follow the precise criteria used to designate an event as lethal or non-lethal. Acknowledge how missing data (e.g., in under-reported regions) might bias the classifier.
- 3.) Chronological Data Splitting – Addressed, but inconsistency remains. While the authors state (Page 19, Paragraph 1) that a geographically stratified chronological split was implemented to prevent temporal leakage, the Appendix (page 22-23) describes a random split for the hyperparameter tuning. I suppose the results in the main text were performed under a chronological split. Still, the random split during hyperparameter search may also indirectly introduce knowledge from the training test into the test set. The choice of the split would significantly affect the validity of conclusions, especially regarding the model's utility for predicting future events.
- 4.) Interpretability and Practical utility – Some progress. I would recommend including the input features that are fed into Random Forest in the Results section, since currently the reader must reach the following section to find out what was used to classify heatwaves. Furthermore, in the rebuttal, the authors mention the risk appetite metric, which I was not able to find. Platt scaling is introduced on page 7, paragraph 2, but is never explained.
- 5.) In general, the manuscript argues that relative humidity and maximum temperature are poor lethality proxies. If we look at Figure 1, we see there is significant overlap between lethal and non-lethal heatwaves, yet the distribution shift among these two populations can still be identified. In this sense, it is surprising that the model trained on max temperature and mean humidity only has such a poor F1 score (Table 1). Did the authors optimize the hyperparameters of the classifier in this case?
- 6.) Overall, there are still poorly written sections with grammar errors or inconsistencies. Please improve the writing. For instance: "...Furthermore, we hope through this classifier better predicting the probability of a heatwave being lethal that it might in turn increase understanding of and responses to ..."

(Remarks on code availability)

Reviewer #2

(Remarks to the Author)

Thanks to the authors for addressing my previous comments

(Remarks on code availability)

Reviewer #3

(Remarks to the Author)

Review- "Reclassifying Lethal Heat" R1

Summary of paper and general comments:

I thank the authors for responding to my comments in the first review. The manuscript has improved, but some of my comments have not been fully addressed in this revised version. I agree that acclimatization and vulnerability are important in understanding the lethality of heatwaves. Yet, the specific application of this global tool in guiding local or national mitigation and adaptation strategies is still not clear. Also, the justification of the predictors chosen for the model is still unclear. My specific comments are included below:

Major comments:

- It is still not clear to me how predicting a binary threshold for lethal/non-lethal heatwaves based on a limited set of meteorological and demographic variables at a global scale is actionable for public health and policy. The binary determination does not provide insight on the level of impact of these heatwave events. Additionally, impacts of heat go beyond mortality and can drive increases in hospitalizations, emergency department visits, etc which are also important to consider from a public health perspective. What are specific actions local, state and federal health departments can implement from the results of this work?
- I thank the authors for adding details on Mora et al., 2017 and their models to identify lethal heat. Yet since both papers classify lethal heat events, it is still not clear to me that the extent to which this paper differentiates from this previously published manuscript. I understand the model is improved with the classifier as compared to using only wet bulb temperature which is consistent with including more variables as inputs to the model. If the novelty is understanding vulnerability and physiological adaptation to lethal heat events, more could be done to consider other socio-demographic variables and vulnerabilities and adaptation.
- This statement on data from lines 179-183, line is misleading - "comprehensive data on underlying comorbidities and physiological adaptation is, for somewhat obvious reasons of practicality, impossible to collect for entire populations within all global subdomains (whether that subdomain is taken to be at the city, county, state, or national level) and is likely to remain so for some time." There are many global socio-demographic data products available at the national, regional and subregional level, some of which were shared in the previous review. If understanding vulnerability and physiological

adaptability to heat is a focus of this manuscript, I think it would be important to consider more socio-demographic variables that are relevant to heat-related health impacts at regional and sub-regional levels in the model framework.

Minor comments:

- I agree that considering temperature differentials is important. However, temperature differentials over 30, 90 or 180 days will likely be capturing seasonal changes in temperature and humidity. In addition to consider seasonal variability, it may be useful to consider metrics of relative heat (days hotter than usual for the season).
- Lines 369-384: "Whilst the data we used was taken from peer reviewed research, setting a relatively high threshold for the quality of the underlying data and meeting the requisite burden of proof, verifying the cause of death and the correct attribution beyond the epidemiological approach already taken within that body of evidence would be extremely challenging in the face of the aforementioned, limited pre- and postmortem diagnosis [55, 56]; more concretely, there is no way of revisiting individual cases and ensuring proper cause of death and attribution when there is likely no body or remaining human tissue of suitable integrity. This is also borne out in the inconsistent reporting on the number of deaths across the data collected by Mora et al [10]. As a result, there is a chance that some heatwaves have been incorrectly labelled and the misclassification of heatwaves remains a possibility; the probability likely varies in accordance with the rate and accuracy of said pre- and postmortem diagnosis and subsequent analysis and is also far more likely to apply to those heatwaves with few fatalities or those incorrectly reported with none at all. A consequence of this is that the model might learn to predict events as nonlethal that were in fact lethal."

It is not clear to me what is meant by these sentences. Many countries have available mortality data that can be used to study heat-related health impacts which are used in the peer-reviewed research used for this study. I would remove these sentences or clarify.

(Remarks on code availability)

I reviewed the GitHub and don't see any major issues although I didn't install and run the code.

Version 2:

Reviewer comments:

Reviewer #1

(Remarks to the Author)

The authors have addressed my main concerns in a satisfactory and constructive manner. The clarification of the data splits and the removal of inconsistencies between the main text and appendix resolve my methodological doubts, and the expanded discussion of biases and limitations in the lethality labels is appropriate and balanced. The additions on feature selection, Platt scaling, and risk thresholds make the method more transparent and easier to interpret for potential users. While some aspects could be further explored in future work, I am now convinced that the analysis is robust and that the manuscript makes a valuable contribution. I support publication in its revised form.

(Remarks on code availability)

31st May 2025

To the reviewers,

REF: Review - Reclassifying Lethal Heat

We appreciate the time and effort that you have dedicated to providing feedback on our manuscript. We have incorporated nearly all of the suggestions made by yourselves and have provided a point-by-point response to each of the comments and concerns in the order that they were presented in the original decision letter, outlined below. All page and paragraph numbers refer to the revised manuscript.

Reviewers' Comments to the Authors:

Reviewer I

1. Binary classification: Provide clear justification for using binary classification.

Author Response: We have provided an expanded justification on the weakness of existing classification using wet bulb temperature thresholds, including citing additional, recent literature, along with discussion on the inconsistent reporting of heatwave excess mortality to motivate this as a better approach from which to begin with classification. [Page 3 Paragraph 3; Page 15 Paragraph 3]

2. Lethality criterion: Provide clear definition of criterion used to provide lethality label, since the task is supervised learning.

Author Response: We have provided a more expansive description of how the lethality label was created. [Page 4 Paragraph 2]

3. Chronological Data Splitting: The current method of randomly splitting the data by years introduces temporal leakage. A chronological train-test split, or a rolling window validation would offer a more realistic evaluation of the model's ability to predict future lethal heatwave events, especially considering climate change effects.

Author Response: Due to the infrequent nature of lethal heatwaves, a pure chronological split affects the representation of some extreme events within the training set; we have created a regional chronological split to counteract this and to address the reviewer's concerns simultaneously. This affects the capability of the model when it comes to predictions in certain parts of Asia with less data for the model to learn from, due to under-reporting of heatwave lethality earlier on in the period of data collection, and have added further commentary on the need for robust data from tropical and under-reporting regions in the next steps. [Page 19 Paragraph 2]

4. Interpretability: The reliance on a Random Forest classifier, limits the utility of the results. The authors should provide clear, actionable insights from the model's predictions and more clear directives on how this model can be used by the practitioners.

Author Response: We have added Platt scaling to the model, a discussion point on tuning the probability thresholds based on risk appetite, and additional commentary on how this research might be used in practice and what it means for public health. [Page 7 Paragraph 2; Page 15 Paragraph 3; Page 16 Paragraphs 3, 4]

5. "Achieves high levels of accuracy":

- The statement on line 322 refers to accuracy even though technically we know that accuracy is not a good measure of how well the algorithm performs for imbalanced classification as stated by the authors elsewhere.
- Suggestion: Consider using terms precision and recall rather than accuracy. A short explanation of what these terms mean can be included.

Author Response: We have amended the text as per this suggestion and in keeping with the focus on FI score (and precision and recall) as throughout the rest of the paper. [Page 16 Paragraph 3]

6. “Two different mechanisms of impact”:

- The phrasing “we believe that there are, in effect, two different mechanisms of impact” is clear, but the explanation following it could benefit from a slight restructuring for clarity. The two mechanisms - (1) exceeding the body’s ability to dissipate heat, and (2) a stress response to rapid environmental changes - are important points, but they could be laid out more succinctly.
- Suggestion: Rephrase as “Our findings suggest two distinct mechanisms for heatwave lethality: one related to the body’s inability to dissipate heat, and the other related to a rapid environmental change triggering a stress response, both of which can prove fatal.” This version is more direct and cohesive.

Author Response: We have amended the text as per this suggestion. [Page 16 Paragraph 3]

7. “Links between climate and public health”:

- The sentence “The strong links between negative outcomes and both climate and public health ought to be of concern” is a bit vague and abstract. It could benefit from more specificity to tie it back to the findings.
- Suggestion: Instead, you could say: “The strong connection between adverse outcomes, climate trends, and public health vulnerabilities is particularly alarming, given the continuing deterioration in both climate and population health.” This strengthens the point and ties it to the context of climate change.

Author Response: We have amended the text as per this suggestion. [Page 16 Paragraph 4]

8. “Force it using future scenarios”:

- The phrase “force it using future climate scenarios and generated meteorological time series” is somewhat awkward. The term “force” might not be the best word choice in this context, where “apply” or “extend” might work better.
- Suggestion: Rephrase as “Our intention is to apply the model to future climate scenarios and generated meteorological time series, along with projections for high-level sociodemographic data.” This sounds smoother and retains the intended meaning.

Author Response: We have amended the text as per this suggestion. [Page 16 Paragraph 4]

9. Final Impact Statement:

- The conclusion about applying the model to future warming scenarios is strong but could be enhanced by tightening the language for clarity.
- Suggestion: “By doing so, we aim to develop a deeper understanding of which regions and populations are most vulnerable to both Shock and Threshold Heatwaves, particularly under high-warming scenarios.” This version is more concise and maintains the same meaning.

Author Response: We have amended the text as per this suggestion. [Page 16 Paragraph 4]

10. Consider rephrasing “fifth generation of global climate reanalysis data, ERA5, and note that is generally considered accurate when”.

Author Response: We have amended the text as per this suggestion. [Page 17 Paragraph 1]

11. Line 363: fix: “out of out of the whole set”.

Author Response: We have amended the text as per this suggestion. [Page 18 Paragraph 2]

12. Section 4.3: it would help to summarise what was the effect of resampling, using the contents of the appendix.

Author Response: We have provided further explanation on the effect of resampling based on the contents of the appendix as suggested in Section 4.3. [Page 19 Paragraph 2]

13. There seems to be missing link and references in the sentence on lines 407-408.

Author Response: This is a placeholder that will be replaced with a zenodo doi for the data and for the code repository, with both being created following the final approval and acceptance of the manuscript and related materials. [Page 20]

14. Since in the end there is no equation to describe the impact of heatwaves but a random forest model it is very important to facilitate the use of this model and clarity of the code by providing concrete explanations in the repository, so consider improving the repository along these lines.

Author Response: The code has been rewritten to create a more flexible and easily applied set of scripts that reproduce the results along with the addition of significant commentary throughout. As per the previous point, if the reviewer deems this acceptable, then a zenodo doi will be created and added into the manuscript. [See GitHub repository]

15. Line 491: “though up to a point” sounds informal, consider rewording. It would help to show that re-sampling does not

qualitatively change the feature distributions.

Author Response: The appendix has been restructured with further explanation and two sets of histogram grid plots for the distribution of each feature in the validation set when compared between the unadjusted dataset and those using either downsampling and synthetic data generation. [Page 22 Paragraph 5; Pages 24, 25]

16. The manuscript appropriately references prior literature but could include more discussion of alternative machine learning models and their applications in similar contexts. A more in-depth comparison of this approach with existing methods for predicting heatwave lethality would provide better context for the contribution. Examples of relevant publications worth discussing: Pir Mohammad, Qihao Weng, Nexus Volume I, Issue 3, 17 September 2024, 100027

Author Response: We thank the reviewer for their recommendation and have cited this, along with additional related literature, in the introduction; one of our key takeaways from this, and the additional literature, was that it served as further evidence of over-reliance on wet bulb temperature variables for classifying lethal heat and believe it strengthens the argument for our work, with the main focus being on the inadequacy of wet bulb temperature variables and which we have further exemplified in an additional experiment that removes it entirely from the model and still achieves the same performance. [Page 3 Paragraph 3; Page 12 Paragraph 3]

Reviewer: 2

Comments to the Author

1. I was struggling to follow the Results section. Could the authors please present the key findings in a more direct and succinct way, e.g., what are the main findings.

Author Response: At the end of the introduction, we have written a paragraph that outlines the key findings and hope that provides suitable description of the main findings and the order in which they appear. [Page 4 Paragraph 1]

2. In the Introduction section, the authors pointed out the struggles that the climate and epidemiology research communities have in defining heatwaves. Please could they clarify why their approach is superior to the approaches used by the climate and epidemiology communities?

Author Response: There was a lack of clarity in our original language that referred to heatwaves rather than, specifically, lethal heatwaves; we have now redressed that and hope that the amended text makes it clear we are not seeking to redefine the meteorological phenomena itself but the definition of whether or not it is likely to be lethal and the struggles previously associated with that. [Page 3 Paragraph 3]

3. The authors used the 90th percentile of temperature distribution as the cut-off to define heatwaves. I would have thought this is similar to the approach used by the epidemiology community. Please could the authors shed some light on this.

Author Response: We have taken this in conjunction with the previous point and believe that the clarification provided around the difference between defining a heatwave and a lethal heatwave makes it clear we are redefining the basis for the latter, not the former. [Page 3 Paragraph 3, Page 4 Paragraph 2]

4. Lines 118-119: In my naïve opinion, one of the main goals of classifying/defining heatwaves is to aid in the development of heatwave response strategies to protect public health. Please could the authors clarify why classifying heatwaves if not to predict their impacts.

Author Response: The reviewer is correct and not stating this explicitly was an oversight that we have now amended with new text discussing that the goal with this framework is to aid in the development of response strategies and to better protect public health. [Page 3 Paragraph 3]

5. I am not too sure if it is the most recommended practice to cite references in the Results section.

Author Response: We understand the reviewer's concern but there is precedent and acceptance for references throughout the results and discussion in articles in Nature Communications across different disciplines (e.g. <https://doi.org/10.1038/s41467-025-59858-0> or <https://doi.org/10.1038/s41467-024-54964-x>) and we feel the references used within those sections of our manuscript are essential for its argument.

6. Please could the authors explain what each variable means in the figure legends (e.g., Figure 5).

Author Response: Within the captions of Figures 1, 4, 5 and Table 1, those that use variable contractions or acronyms, we have

provided a definition. [Pages 5, 10, 11, 12, 24, and 25]

Reviewer 3

1. The utility of this classification algorithm and relevance to policy or public health is not clear. Along the same line, using a binary lethal/non-lethal classification without considering the impact such as the number of deaths is not well justified.

Author Response: We have written a clearer justification based on the fact that current decision boundaries for deciding whether or not a heatwave is going to be lethal, including research published as recently as 2024, still use wet bulb temperature thresholds. To help underscore the inappropriateness of this, we have included a model run where the instantaneous temperature and humidity were not used as inputs at all and the impact to model performance is negligible. We have also included an expanded section in the discussion about this work's real world impact, how it affects public health and policy, and where we intend to take the research next. [Page 3 Paragraphs 2, 3; Page 12 Paragraph 3; Page 15 Paragraph 3; Page 16 Paragraph 3]

2. The authors state that "Epidemiological studies focus mostly on total population mortality and morbidity outcomes without differentiating between socially vulnerable groups." This is overlooking an abundance of research evaluating social vulnerability to extreme heat from epidemiology. A few relevant articles include:

- Son, J. Y., Liu, J. C., & Bell, M. L. (2019). Temperature-related mortality: a systematic review and investigation of effect modifiers. *Environmental Research Letters*, 14(7), 073004.
- Benmarhnia, T., Deguen, S., Kaufman, J. S., & Smargiassi, A. (2015). Vulnerability to heat-related mortality: A systematic review, meta-analysis, and meta-regression analysis. *Epidemiology*, 26(6), 781-793.
- Arsad, F. S., Hod, R., Ahmad, N., Ismail, R., Mohamed, N., Baharom, M., ... & Tangang, F. (2022). The impact of heatwaves on mortality and morbidity and the associated vulnerability factors: a systematic review. *International Journal of Environmental Research and Public Health*, 19(23), 16356.
- Bakhtsiyarava, M., Schinasi, L. H., Sánchez, B. N., Dronova, I., Kephart, J. L., Ju, Y., ... & Rodríguez, D. A. (2023). Modification of temperature-related human mortality by area-level socioeconomic and demographic characteristics in Latin American cities. *Social Science & Medicine*, 317, 115526.

Author Response: We would like to thank the reviewer for their recommendation and have included all recommended literature along with some additions within the introduction in an updated paragraph. [Page 2 Paragraph 2]

3. The manuscript uses data from previous work (Mora et al., 2017), and more substantial description of the published manuscript and what this new study adds would be important. The added value of this particular classification tool should be clarified.

Author Response: We have provided the requested summarisation in the introduction and have further cited this prior art where appropriate. [Page 3 Paragraph 3]

4. The manuscript states: "Consequently, heat-related mortality data are sparse and have not been analyzed consistently [6, 7, 9]" The citations listed are outdated, and there is extensive literature on this topic that has not been cited. Here are a few:

- Ebi, K. L., Capon, A., Berry, P., Broderick, C., de Dear, R., Havenith, G., ... & Jay, O. (2021). Hot weather and heat extremes: health risks. *The Lancet*, 398(10301), 698-708.
- Khatana, S. A. M., Werner, R. M., & Groeneveld, P. W. (2022). Association of extreme heat with all-cause mortality in the contiguous US, 2008-2017. *JAMA Network Open*, 5(5), e2212957-e2212957.
- Kephart, J. L., Sánchez, B. N., Moore, J., Schinasi, L. H., Bakhtsiyarava, M., Ju, Y., ... & Rodríguez, D. A. (2022). City-level impact of extreme temperatures and mortality in Latin America. *Nature Medicine*, 28(8), 1700-1705.
- Rocklöv, J., Ebi, K., & Forsberg, B. (2011). Mortality related to temperature and persistent extreme temperatures—a study of cause-specific and age stratified mortality. *Epidemiology*, 22(1), S13.

Author Response: We would like to thank the reviewer for their recommendation; the text has been updated and restructured to include references to the additional recommended literature. [Page 3 Paragraphs 2, 3]

5. A more detailed description of how the heatwaves were labelled as lethal or nonlethal should be included. Is there a possibility of misclassification in this categorization? How can we be sure that the heatwaves categorized as non-lethal heatwaves did not cause any deaths and is there a way to validate this?

Author Response: As per a similar point raised by Reviewer 1, we have provided a more expanded description of how the lethality label was created. There certainly is a chance of misclassification and we have added in a paragraph describing this risk, the fact that it is challenging to mitigate, and the associated implications. [Page 4 Paragraph 2; Page 15 Paragraph 3]

6. Physiological vulnerability to extreme heat encompasses a range of health factors including pre-existing health conditions and socio-demographics. Relying on BMI as an indicator of population health in relation to heat is unlikely to fully capture this

vulnerability. Additionally, BMI is a flawed indicator of health (see Ahima, R. S., & Lazar, M. A. (2013). The health risk of obesity—better metrics imperative. *Science*, 341(6148), 856-858.) The manuscript states that “comprehensive data on underlying comorbidities and physiological adaptation is, for somewhat obvious reasons of practicality, impossible to collect for entire populations within all global subdomains (whether that subdomain is taken to be at the city, county, state, or national level) and is likely to remain so for some time”, but there are many potential data sources that could be used that will have information beyond age and BMI. Some examples include:

- Global Burden of Disease study: <https://ghdx.healthdata.org/series/global-burden-disease-gbd>
- Global socio-demographic index (SDI): <https://ghdx.healthdata.org/record/global-burden-disease-study-2021-gbd-2021-socio-demographic-index-sdi-I950%E2%80%932021>
- GLOPOP-S: <https://www.nature.com/articles/s41597-024-03864-2>

Author Response: We would like to thank the reviewer for their suggestion; unfortunately, many of the suggestions and related datasets do not fully overlap with the period of study in this paper, which prevents their usage. One counter example, however, is that of the global socio-demographic index; we have incorporated this dataset into the study and paper and added commentary around its usage and impact. We have also acknowledged within the paper that BMI is an imperfect metric but the model’s strength when compared to pre-existing wet bulb temperature models is quite reliant on BMI, indicating that, although imperfect, the information it contains on a population’s vulnerability to heat is vital to prediction. [Page 6 Paragraph 3; Page 12 Paragraph 2]

7. As there are some false negative and false positive predictions, a discussion of the potential consequences in the misclassification of the lethality of heatwaves would be important.

Author Response: We have added a discussion of the consequences of misclassification as requested. [Page 15 Paragraph 3]

8. The discussion of what data is being used to train this model and what regions are missing from the data used would be important to contextualize the results.

Author Response: Further information has been included within the discussion to help provide said contextualisation, pointing to the absence of data from the Global South, the implications of that missing data, and what additional variables might be required to generalise the model were this data to exist for the period of study and moving forward. [Page 14 Paragraph 3; Page 15 Paragraph 2]

9. Abstract: “Furthermore, we found that the majority of level heatwaves within our dataset..”- is this a typo?

Author Response: This was indeed a typographical error that we have now corrected. [Page 2]

10. Although the severity of these events varies in terms of the total mortality, such as 4,867 excess deaths in Paris arising from the 2003 heatwave [17] and 11 for the 1999 heatwave that affected Minneapolis [18]...” This does not reference a study conducted in Minneapolis.

Author Response: Thank you for spotting this error mixing up 2 US cities; the text has been amended to correctly refer to Milwaukee. [Page 4 Paragraph 2]

11. The GitHub and README file have several placeholders and portions that are not finalized that should be edited.

Author Response: Please see our response to point 14 made by reviewer I, which is of a very similar vein, and which we hope has been addressed fully. [See GitHub repository.]

Yours Sincerely,

Robert Edwin Rouse

Department of Applied Mathematics and Theoretical
Physics, University of Cambridge, Cambridge, UK
CB3 0WA

David Andrew Rouse

Retired Home Office Forensic Pathologist,
The Home Office, London, UK
SW1P 4DF

Ramit Debnath

Department of Architecture,
University of Cambridge, Cambridge, UK
CB2 1PX

Will Tebutt

Department of Engineering,
University of Cambridge, Cambridge, UK
CB2 1PZ

Bénédicte Dousset

Department of Geography and Environment,
University of Hawai‘i at Manoa, Honolulu
US, HI 96822

Camilo Mora

Department of Geography and Environment,
University of Hawai‘i at Manoa, Honolulu
US, HI 96822

Scott Hosking

British Antarctic Survey, High Cross,
Cambridge, UK
CB3 0ET

Allan McRobie

Department of Engineering,
University of Cambridge, Cambridge, UK
CB2 1PZ

Emily Shuckburgh

Department of Computer Science,
University of Cambridge, Cambridge, UK
CB3 0FD

10th October 2025

To the reviewers,

REF: Review - Reclassifying Lethal Heat

We appreciate the time and effort that you have dedicated to providing further feedback on our manuscript. We have provided a point-by-point response to each of the comments and concerns in the order that they were presented in the second decision letter, outlined below. All page and paragraph numbers refer to the revised manuscript.

Reviewers' Comments to the Authors:

Reviewer I

1. Definition of Lethality Labels – Partially Addressed. While the authors have provided a more expansive description of lethality labelling, it remains difficult to follow the precise criteria used to designate an event as lethal or non-lethal. Acknowledge how missing data (e.g., in under-reported regions) might bias the classifier.

Author Response: We appreciate the reviewer's comment and have streamlined the text to improve clarity and accessibility. As to the second half of this point, we have discussed at length how missing data or incorrectly reported data might bias the classifier in the discussion section. [Page 3 Paragraphs 3, 4]

2. Chronological Data Splitting – Addressed, but inconsistency remains. While the authors state (Page 19, Paragraph 1) that a geographically stratified chronological split was implemented to prevent temporal leakage, the Appendix (page 22-23) describes a random split for the hyperparameter tuning. I suppose the results in the main text were performed under a chronological split. Still, the random split during hyperparameter search may also indirectly introduce knowledge from the training test into the test set. The choice of the split would significantly affect the validity of conclusions, especially regarding the model's utility for predicting future events.

Author Response: The data used in the validation set was taken randomly from the training set after the chronological split was made for the test set, such that no data from the test set has been seen by the model at any stage except for the generation of final results and no knowledge was introduced into the training set from the test set. We have updated the text for added clarity, explicitly stating that the data has been partitioned into separate sets with 80% in the training set, 10% in the validation set, and 10% in the test set with no overlaps between them. [Page 20 Paragraph 1]

3. Interpretability and Practical utility – Some progress. I would recommend including the input features that are fed into Random Forest in the Results section, since currently the reader must reach the following section to find out what was used to classify heatwaves. Furthermore, in the rebuttal, the authors mention the risk appetite metric, which I was not able to find. Platt scaling is introduced on page 7, paragraph 2, but is never explained.

Author Response: We have added a paragraph toward the beginning of the results section that lists all input variables. As for the risk appetite and probability threshold, we agree with the reviewer that further explanation was required and have now included this both for adjusting the probability threshold and on the use of Platt Scaling. [Page 7 Paragraph 1; Page 7 Paragraph 2]

4. In general, the manuscript argues that relative humidity and maximum temperature are poor lethality proxies. If we look at Figure 1, we see there is significant overlap between lethal and non-lethal heatwaves, yet the distribution shift among these two populations can still be identified. In this sense, it is surprising that the model trained on max temperature and mean humidity only has such a poor FI score (Table 1). Did the authors optimize the hyperparameters of the classifier in this case?

Author Response: We appreciate the reviewer's viewpoint and have added further text within the relevant section to further

underscore the point that the probability of a heatwave being lethal given its exceeding a wet bulb temperature threshold of either 25°C and 30°C is very low and helps to make the result less surprising. [Page 5 Paragraphs 1 & 2]

5. Overall, there are still poorly written sections with grammar errors or inconsistencies. Please improve the writing. For instance: "...Furthermore, we hope through this classifier better predicting the probability of a heatwave being lethal that it might in turn increase understanding of and responses to ...".

Author Response: We have rewritten multiple sections in the manuscript, simplifying the language to improve readability.

Reviewer: 2

We thank the reviewer for their time in reviewing this submission and for their constructive criticism.

Reviewer 3

1. It is still not clear to me how predicting a binary threshold for lethal/non-lethal heatwaves based on a limited set of meteorological and demographic variables at a global scale is actionable for public health and policy. The binary determination does not provide insight on the level of impact of these heatwave events. Additionally, impacts of heat go beyond mortality and can drive increases in hospitalizations, emergency department visits, etc which are also important to consider from a public health perspective. What are specific actions local, state and federal health departments can implement from the results of this work?

Author Response: We thank the reviewer for their point and have now pointed out in this manuscript that the reliance on wet bulb temperature thresholds or just temperature thresholds persists to date and is the prevalent method of identifying lethal heat conditions, resulting in severe under-prediction of lethal heat conditions. For example, we explicitly cite the use of absolute thresholds even in countries with more advanced meteorological and health security agencies, such as the UK Health Security Agency and Met Office in the United Kingdom relying on absolute thresholds of just temperature, not even wet bulb temperature, for their early warning systems. As we have shown, our model is a significant improvement and can be applied globally to identify a more accurate decision boundary subject to local conditions and history therefore could enable state and/or regional actors to identify lethal conditions for a given population. We do agree that the specific design of policy and/or mitigation strategies is an interesting downstream activity but one that falls outside the scope of the current study, which focuses on the development of a superior modelling framework for identifying lethal conditions. [Page 3 Paragraph 3]

2. I thank the authors for adding details on Mora et al., 2017 and their models to identify lethal heat. Yet since both papers classify lethal heat events, it is still not clear to me that the extent to which this paper differentiates from this previously published manuscript. I understand the model is improved with the classifier as compared to using only wet bulb temperature which is consistent with including more variables as inputs to the model. If the novelty is understanding vulnerability and physiological adaptation to lethal heat events, more could be done to consider other socio-demographic variables and vulnerabilities and adaptation.

Author Response: We do agree with the reviewer's viewpoint that including additional other socio-demographic variables would be useful but, due to the issues with incompatible spatio-temporal overlap, we cannot use many that we might wish to. We have provided further text to underline this point within the manuscript. As to the novelty, the manuscript cites recent papers, some of which are even from this year, on the use of wet bulb temperatures as a classifier of lethal heat persisting both in practice and in literature. We have shown that it is not an adequate classifier and our model is at least an order of magnitude more accurate, when adjusting class imbalance, which is a significant improvement. Furthermore, it is not the case that simply adding more variables improves the accuracy; we have shown that the absolute wet bulb temperature conditions for many countries outside of the tropics can effectively be ignored altogether. [Page 6 Paragraph 2]

3. This statement on data from lines 179-183, line is misleading - "comprehensive data on underlying comorbidities and physiological adaptation is, for somewhat obvious reasons of practicality, impossible to collect for entire populations within all global subdomains (whether that subdomain is taken to be at the city, county, state, or national level) and is likely to remain so for some time." There are many global socio-demographic data products available at the national, regional and subregional level, some of which were shared in the previous review. If understanding vulnerability and physiological adaptability to heat is a focus of this manuscript, I think it would be important to consider more socio-demographic variables that are relevant to heat-related health impacts at regional and sub-regional levels in the model framework.

Author Response: We thank the reviewer for their comment and believe that our point is aligned with their viewpoint as we discuss the need for socio-demographic variables as proxies due to the comprehensive diagnosis of all underlying comorbidities within a

population being impossible. We have amended the text to improve clarity but, as discussed in the response to the previous point, any datasets used must have full spatio-temporal overlap with the labelled data, which has restricted usage to specific variables as described in the paper. [Page 5 Paragraph 2; Page 6 Paragraph 2]

4. I agree that considering temperature differentials is important. However, temperature differentials over 30, 90 or 180 days will likely be capturing seasonal changes in temperature and humidity. In addition to consider seasonal variability, it may be useful to consider metrics of relative heat (days hotter than usual for the season).

Author Response: We appreciate the reviewer's insight and agree this is useful information; however, the formulation of the temperature differentials, in terms of differential over averages of different temporal ranges, was designed explicitly to capture the effects of hotter than average temperatures preceding the heatwave, affecting some acclimatisation to the conditions of the heatwave, in addition to the seasonal changes and local variability between seasons. We have added some additional text to describe this explicitly. [Page 6 Paragraph 3]

5. Lines 369-384: "Whilst the data we used was taken from peer reviewed research, setting a relatively high threshold for the quality of the underlying data and meeting the requisite burden of proof, verifying the cause of death and the correct attribution beyond the epidemiological approach already taken within that body of evidence would be extremely challenging in the face of the aforementioned, limited pre- and postmortem diagnosis [55, 56]; more concretely, there is no way of revisiting individual cases and ensuring proper cause of death and attribution when there is likely no body or remaining human tissue of suitable integrity. This is also borne out in the inconsistent reporting on the number of deaths across the data collected by Mora et al [10]. As a result, there is a chance that some heatwaves have been incorrectly labelled and the misclassification of heatwaves remains a possibility; the probability likely varies in accordance with the rate and accuracy of said pre- and postmortem diagnosis and subsequent analysis and is also far more likely to apply to those heatwaves with few fatalities or those incorrectly reported with none at all. A consequence of this is that the model might learn to predict events as nonlethal that were in fact lethal." It is not clear to me what is meant by these sentences. Many countries have available mortality data that can be used to study heat-related health impacts which are used in the peer-reviewed research used for this study. I would remove these sentences or clarify.

Author Response: We have altered the text to provide further clarity and cited an additional study that shows global autopsy and post mortem diagnosis rates are falling worldwide, rather than increasing, underscoring the challenge of correct attribution when fatality rates are within typical noise levels. [Page 16 Paragraph 3]

Yours Sincerely,

Robert Edwin Rouse

Department of Applied Mathematics and Theoretical
Physics, University of Cambridge, Cambridge, UK
CB3 0WA

David Andrew Rouse

Retired Home Office Forensic Pathologist,
The Home Office, London, UK
SW1P 4DF

Ramit Debnath

Department of Architecture,
University of Cambridge, Cambridge, UK
CB2 1PX

Will Tebutt

Department of Engineering,
University of Cambridge, Cambridge, UK
CB2 1PZ

Bénédicte Dousset

Department of Geography and Environment,
University of Hawai'i at Manoa, Honolulu
US, HI 96822

Camilo Mora

Department of Geography and Environment,
University of Hawai'i at Manoa, Honolulu
US, HI 96822

Scott Hosking

British Antarctic Survey, High Cross,
Cambridge, UK
CB3 0ET

Allan McRobie

Department of Engineering,
University of Cambridge, Cambridge, UK
CB2 1PZ

Emily Shuckburgh

Department of Computer Science,
University of Cambridge, Cambridge, UK
CB3 0FD